# Estimating lateral nitrogen transfer through the global river network using a land surface model

Minna Ma[1*], Haicheng Zhang[2*], Ronny Lauerwald[3], Philippe Ciais[4], Pierre Regnier[1]

[1] Department Geoscience, Environment & Society-BGEOSYS, Université libre de Bruxelles, 1050 Bruxelles, Belgium

[2] School of Geography and Planning, Sun Yat-sen University, Guangzhou, Guangdong, 510006, China

[3] Université Paris-Saclay, INRAE, AgroParisTech, UMR ECOSYS, Palaiseau, France

[4] Laboratoire des Sciences du Climat et de l' Environnement, IPSL-LSCE CEA/CNRS/UVSQ, Orme des Merisiers, 91191, Gif sur Yvette, France

Correspondence: Minna Ma (minna.ma@ulb.be) and Haicheng Zhang (zhanghch59@mail.sysu.edu.cn).

**Abstract.** Lateral nitrogen (N) transport from land to oceans through rivers is an important component of the global N cycle. We developed a new model of this system, called ORCHIDEE-NLAT, which simulates the routing of water in rivers, and the pertaining transport of dissolved inorganic N (DIN), dissolved organic N (DON) and particulate organic N (PON) as well as the accompanying biogeochemical processes of decomposition for DON and PON, and denitrification for DIN during the transit from land to oceans through the river network. Evaluation against global observation-based datasets reveal that the model captures both the magnitude and seasonal variations of riverine water discharges and total nitrogen (TN) flows well. The ORCHIDEE-NLAT model was then applied to reconstruct the historical evolution of global TN flows from land to rivers, as well as the denitrification of DIN within the river network. Due to anthropogenic activities (e.g. mineral fertilisers and manure application, sewage water injection in rivers and land use change) and indirect climate and $CO_2$ effects, the TN exports are modelled to increase from 27.1 Tg N yr$^{-1}$ over 1901-1910 to 40.8 Tg N yr$^{-1}$ over 2001-2014, with DIN (80%) contributing most of this increase. The annual mean TN flow and DIN denitrification rates show substantial spatial heterogeneities. The seasonal amplitude of TN flow is of similar magnitude as the large-scale spatial variability. Compared to previously published global aquatic N transfer models (IMAGE-GNM, FrAMES-N, MBM, DLEM and Global NEWS2), our model produces similar global and continental-scale TN exports to the ocean, but shows distinct patterns at finer scale spatial scales (e.g. basin scale). ORCHIDEE-NLAT could also be coupled with other land surface models such as those used in the Nitrogen Model Intercomparison Project (NMIP). Our model provides a full simulation of N transport and reactivity from soils to oceans at an unprecedented spatio-temporal resolution (daily fluxes at 0.5° globally).

## 1. Introduction

Nitrogen (N) is an essential element for all life on Earth, and the N cycle interacts in multiple ways with the Earths climate system and the environment. Nitrous oxide ($N_2O$) is an important greenhouse gas, which affects Earths energy balance in a similar way as carbon dioxide ($CO_2$) but is nearly 300 times more effective on a per molecule basis (Sainju et al., 2014). N also affects the C cycling, $CO_2$ and methane ($CH_4$) fluxes as it limits primary production rates in many terrestrial, freshwater and marine ecosystems (Thornton et al., 2007; Moore et al., 2013; Zaehle et al., 2014; Seiler et al., 2024). The N cycle thus plays an important role in controlling the C cycle and climate change, which calls for an analysis of the N dynamics in the context of changing C cycle, climate and anthropogenic activities. In this Earth system perspective, insufficient attention has been given to the tight link between the terrestrial and marine N cycles through the Land to Ocean Aquatic Continuum (LOAC) (Galloway et al., 2003; Billen et al., 2013; Maranger et al., 2018). Existing studies have largely treated the land and open ocean cycles separately, ignoring N processes occurring along the LOAC (Fowler et al., 2013; Zhang et al., 2021). The representation of N processes in the LOAC is required to achieve a dynamic coupling between land surface and ocean biogeochemical models, with land surface models simulating the dynamically changing N exports to the coast, which may include historical hindcasts and future projections.

Over the past several decades, the cumulative effects of climate change, increased population, industrialization and agricultural fertiliser use have accelerated the global N cycle, and increased N leaching into the LOAC (Bouwman et al., 2005; Kim et al., 2011; Swaney et al., 2012; Beusen et al., 2016). This has resulted in negative human health and environmental impacts such as drinking water degradation and an increase in frequency and severity of eutrophication (Dodds & Smith, 2016; Huang et al., 2017; Costa et al., 2018; Lee et al., 2019; Dai et al., 2023). Most land surface models include N leaching

to aquatic systems, this process is rarely evaluated in quantitative terms using observations collected within the fluvial network. It has been shown that N leaching is inaccurate in most LSMs (Feng et al., 2023), which in turn affects the simulation of the response of terrestrial C and N cycles to anthropogenic activities and climate change (Thomas et al., 2013). Furthermore, explicit representation of the fate of the land-derived N inputs into the LOAC is required to better constrain the response of the ocean C cycle to increased nutrient inputs (Lacroix et al., 2021; Resplandy et al., 2024) as well as to assess the extent to which N pollution reduction scenarios can mitigate (Satter et al., 2014) eutrophication in riverine and coastal aquatic ecosystems (Hashemi et al., 2016; Desmit et al., 2018).

The representation of N lateral transfers through aquatic systems is challenging because it requires to represent multiple N sources, transformation, transport, and retention processes along the global fluvial network. A variety of models with different structures and representations of the water and N cycles have been developed  (Luscz et al., 2015, 2017). Models such as SWAT (the Soil and Water Assessment Tool) (Arnold et al., 1998; Liu et al., 2017), HSPF (the Hydrologic Simulation Program-FORTRAN) (Bicknell et al., 2005; Wang et al., 2015) and HYPE (HYdrological Predictions for the Environment) (Lindstrom et al., 2010; Donnelly et al., 2014) represent hydrology and N transport and transformation processes in rivers, but mainly for catchment scale applications. Therefore, their complexity and high data requirements for calibration and evaluation limit their applicability, in particular the long-term evolution of global N fluxes and transformation processes. Simplified empirical approaches provide an alternative for large-scale simulations, such as the Global NEWS2 model (Global Nutrient Export from Watersheds 2) allowing to estimate riverine N exports to the ocean as a function of N deliveries from the surrounding catchment with a highly simplified representation of N transport

and in-stream N processes (Seitzinger et al., 2005; Mayorga et al., 2010; Lee et al., 2016). The Integrated Model to Assess the Global Environment-Global Nutrient Model (IMAGE-GNM) provides a more process-based representation of the river network as it relies on a globally distributed, spatially explicit hydrological model (PCR-GLOBWB, PCR aster Global Water Balance) to estimate N delivery to surface waters and its subsequent transport (Beusen et al. 2014, 2016 & 2022; Vilmin et al., 2018). This model however simulates N retention according to empirical formulas, is not dynamically coupled to vegetation-soil N processes and only provides yearly averaged fluxes, hence ignoring the seasonal fluctuations induced by the hydrology and N cycling on land and in the river network. The Dynamic Land Ecosystem Model (DLEM 2.0) was improved to simulate riverine N flow from terrestrial ecosystems to rivers and coastal oceans. So far, however, the N lateral transfer simulated by DLEM has only been evaluated at regional scale (eastern North America, Yang et al., 2015) or for $N_2O$ emissions on the global scale (Tian et al. 2018; Yao et al., 2020). To complement these studies, we develop a new N lateral transfer model that can be linked to the outputs of land surface models while capturing the hydrology and N transformation processes in the global river network at a temporal resolution (days to months) as relevant for biogeochemical processes in coastal and marine ecosystems. At the same time, this model should be able to reconstruct and forecast the long-term (decadal to century-scale) evolution of the aquatic N cycle as a result of a wide variety of anthropogenic factors, including climate change.

Our model is an offline model of lateral N transfers which is fed with outputs from the land surface model ORCHIDEE. ORCHIDEE is a widely used land-surface model (Krinner et al., 2005), with many versions (or branches) focusing on different aspects of the terrestrial C cycle and associated bio-elements. We leverage ORCHIDEE-CNP, the branch simulating the coupled

cycles of carbon (C), N and phosphorus (P) in the terrestrial biosphere (Sun et al., 2021), and ORCHIDEE-Clateral, the branch simulating the leaching and erosion of C along the soil-inland water continuum (Lauerwald et al., 2017; Lauerwald et al., 2020; Zhang et al., 2022). Our study is structured as follows: (1) development of an offline N lateral transfer model (ORCHIDEE-NLAT) driven by output from ORCHIDEE-Clateral and ORCHIDEE-CNP; (2) collection of observations of water discharge and N concentration to evaluate the performance of ORCHIDEE-NLAT; (3) investigation of the spatio-temporal dynamics of N lateral transfer over the historical period (1900-2014); and (4) comparison of model results with those obtained with previously published models.

# 2. Methods and Data

## 2.1. Model development

### 2.1.1. The ORCHIDEE-NLAT model

The ORCHIDEE land surface model comprehensively simulates the cycling of energy, water and C, in terrestrial ecosystems (Krinner et al., 2005). As the model evolved, many versions (or branches) emerged with various foci on additional land surface processes impacting the climate system. In particular, the ORCHIDEE-CNP branch features a detailed representation of the coupled cycling of C, N, and P in vegetation and soil (e.g. root uptake of N, the allocation of N in the tissue of different parts of vegetation biomass, N turnover in litter and soil organic matter) and the leaching of $NH_4^+$ and $NO_3^-$ from soils to inland waters (Goll et al., 2017, 2018; Sun et al., 2021). The ORCHIDEE-Clateral branch stimulates the large-scale lateral transfer and fate of water, sediment, particulate (POC) and dissolved organic C (DOC), and $CO_2$ along the land-river-ocean continuum (Lauerwald et al., 2017; Zhang et al., 2020, 2022).

Based on the land-to-river inputs of water, POC, DOC and inorganic N simulated by ORCHIDEE-CNP and ORCHIDEE-Clateral, we developed the ORCHIDEE-NLAT model to simulate the transfers of reactive N through the global river network. We use an offline approach which has the advantage of running fast, and the potential to be coupled with output from other LSMs. In this offline approach, ORCHIDEE-CNP provides as input the leaching rates of terrestrial dissolved inorganic N (DIN) with surface runoff and subsoil drainage and dissolved organic N (DON) leaching from manure. Inputs of terrestrial DON and particulate organic N (PON) are derived from the leaching and erosional fluxes of DOC and POC simulated by ORCHIDEE-Clateral and stoichiometric C:N ratios of dissolved organic matter (DOM) and particulate organic matter (POM), please refer to section 2.1.2 for details (Fig. 1).

During the twentieth century, global N (DIN and DON) discharge to surface water from sewage increased about 3.5-fold to 7.7 Tg N $yr^{-1}$, which has large impact on the global N lateral transfer. N discharge from sewage also included in ORCHIDEE-NLAT using N sewage dataset (1900-2010, gridded maps every five years) reported by Beusen et al. (2016). N in sewage comes from three kinds of sources: human waste from urban environments, animal waste, and industrial waste, which has different fates, please read details in Van Drecht (2009) and Morée et al (2013).

PON, DON and DIN are transported by advection with the flow of water: from soils to rivers and through the river network all the way to the coast. Within the river network, part of the transported DON and PON is decomposed to DIN, and part of DIN is released back to the atmosphere through denitrification processes. Following previous global modelling approaches (Aitkenhead-Peterson et al., 2001; Bernot and Dodds, 2005; Wollheim et al., 2008), ORCHIDEE-NLAT simulates DIN denitrification without explicit

representation of the different DIN species (i.e. $NO_3^-$ and $NH_4^+$) or their interconversion via nitrification (Fig.1).

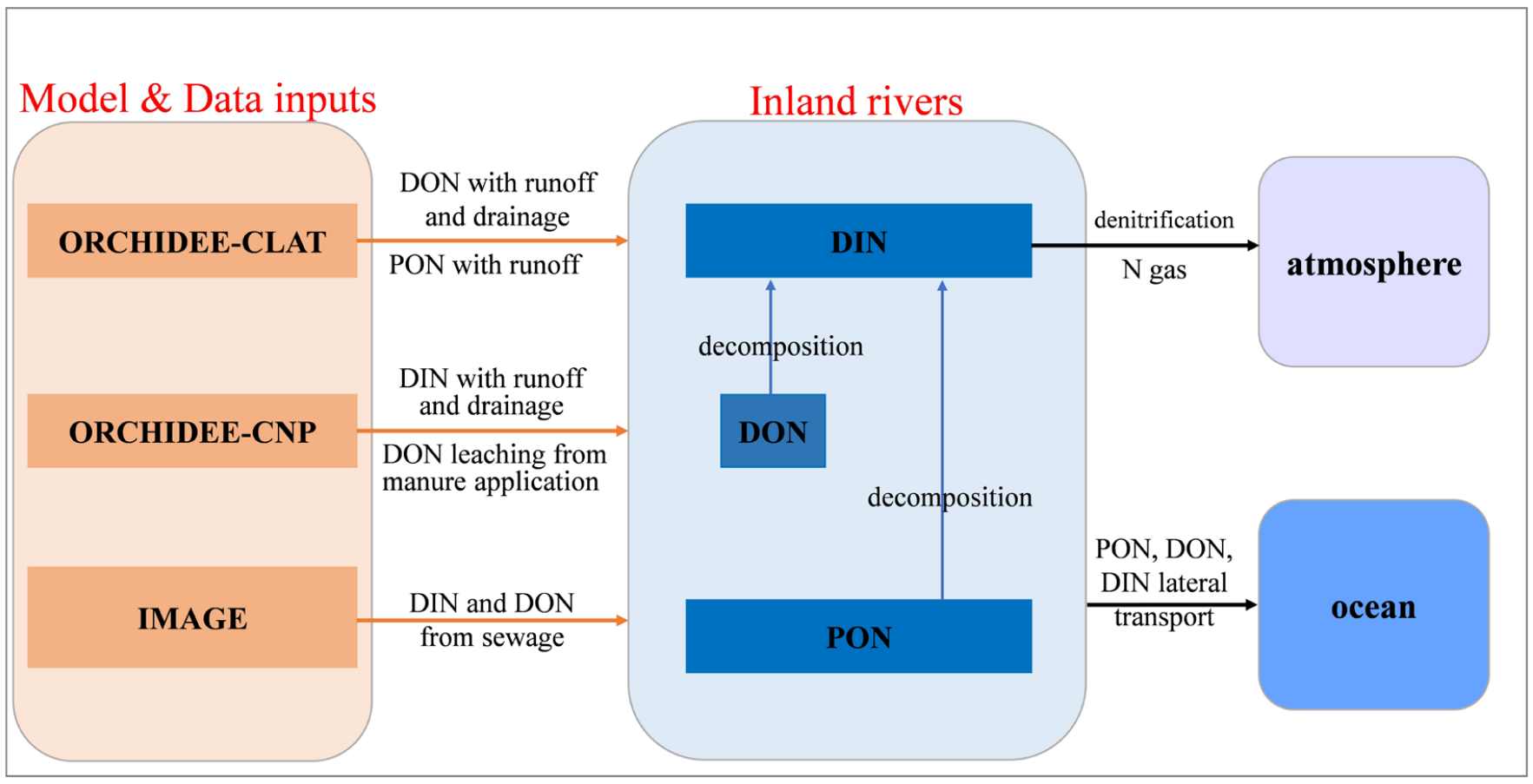

Figure 1. Sources of themodel driving data and the main aquatic N transformation processes in ORCHIDEE-NLAT.

### 2.1.2. N delivery from upland soils to the river network

The lateral transfer of DOC and POC from land to rivers was used to constrain DON and PON inputs. PON erosion with runoff originates from three soil organic matter (SOM) pools characterised by different C:N ratios set to 12, 25 and 8 for active, slow and passive SOM pools, respectively (Zhang et al., 2022). The PON erosion from each pool is calculated by dividing the POC erosion flux from the same SOM pool by its corresponding C:N ratio. For DON leaching with runoff and drainage, we relied on measurements of the stoichiometry of dissolved organic matter, which report C:N ratios in soil and rivers comprised between 8 and 25, with an average value of around 12 (Kirkby et al., 2011; Lutz et al., 2011; Tipping et al., 2016; Maranger et al., 2018; Rodríguez-Cardona et al., 2021). Therefore, the leaching of DON with runoff and drainage were quantified from ORCHIDEE-Clateral simulations of the corresponding DOC fluxes and an average C:N ratio of 12, noting that the resulting flow excludes the DON leaching caused by manure application (this

source is not accounted for in ORCHIDEE-Clateral). The spatial and temporal resolution of the resulting DON and PON flow used to force ORCHIDEE-NLAT was 1° with a timestep of one day (Table 1) and these inputs were resampled to the nominal resolution of ORCHIDEE-NLAT of 0.5° using the nearest-neighbour resampling (Patil, 2018).

DIN (i.e. $NH_4^+$ and $NO_3^-$) inputs from soils to rivers was prescribed from a simulation of ORCHIDEE-CNP (Goll et al., 2017a, 2018; Sun et al., 2021) which include DIN leaching from natural and cultivated (e.g. cropland and pasture) ecosystems, and account for changes induced by atmospheric N deposition, fertiliser use and manure application. DON inputs to rivers from manure application were prescribed from ORCHIDEE-CNP based on a DON pool and leaching factor, a separate DON pool from manure being added into ORCHIDEE-CNP to participate in the subsequent N cycling and leaching processes. The spatial and temporal resolution of this input dataset was 2° with a daily time step and were downscaled to the ORCHIDEE-NLAT spatial resolution of 0.5° using the nearest-neighbour resampling (Patil, 2018) (Table 1).

Finally, the N inputs from sewage (https://doi.org/10.17026/dans-zgs-9k9m) provided at 0.5° globally but with a yearly timestep (Beusen et al, 2016) were redistributed evenly across each day of the year (Table 1).

### 2.1.3. N transport and transformation in the river network

ORCHIDEE-NLAT simulates river discharge along a distributed routing scheme (Vörösmarty et al., 2000). As shown in Fig. 2, surface runoff ($F_{RO}$) and belowground drainage ($F_{DR}$), both as model inputs extracted from ORCHIDEE-Clateral, first feed into the "fast" ($S_{fast_H2O}$, $m^3$) and "slow" water reservoirs ($S_{slow_H2O}$, $m^3$), respectively. The delayed outflows from these reservoirs then feed into the "stream" water reservoir ($S_{stream_H2O}$, $m^3$). The outflow rates from

the fast ($F_{fastout_H2O}$, $m^3$ $d^{-1}$) and slow ($F_{slowout_H2O}$, $m^3$ $d^{-1}$) reservoirs are calculated at a daily time-step based on a grid-cell-specific topographic index $f_{topo}$ (unitless, Vörösmarty et al., 2000) (Table 1) and a reservoir-specific water turnover factor $\tau$, which translates $f_{topo}$ into a water residence time for each reservoir attached to each river segment (Eqs. 1 and 2). Water in the stream reservoir ($S_{stream_H2O}$) in grid cell $i$ then flows downstream (Eq. 3) into the stream reservoir of grid cell $i+1$ ($F_{downstream_H2O}$, $m^3$ $d^{-1}$). The $\tau_{fast}$, $\tau_{slow}$ and $\tau_{stream}$ are set to 3.0 days, 25.0 days and 0.24 days, which are default settings in ORCHIDEE (Ngo-Duc et al., 2006).

$$F_{fastout_H2O} = \frac{S_{fast_H2O}}{\tau_{fast} \times f_{topo}} \quad (1)$$

$$F_{slowout_H2O} = \frac{S_{slow_H2O}}{\tau_{slow} \times f_{topo}} \quad (2)$$

$$F_{downstream_H2O} = \frac{S_{stream_H2O}}{\tau_{stream} \times f_{topo}} \quad (3)$$

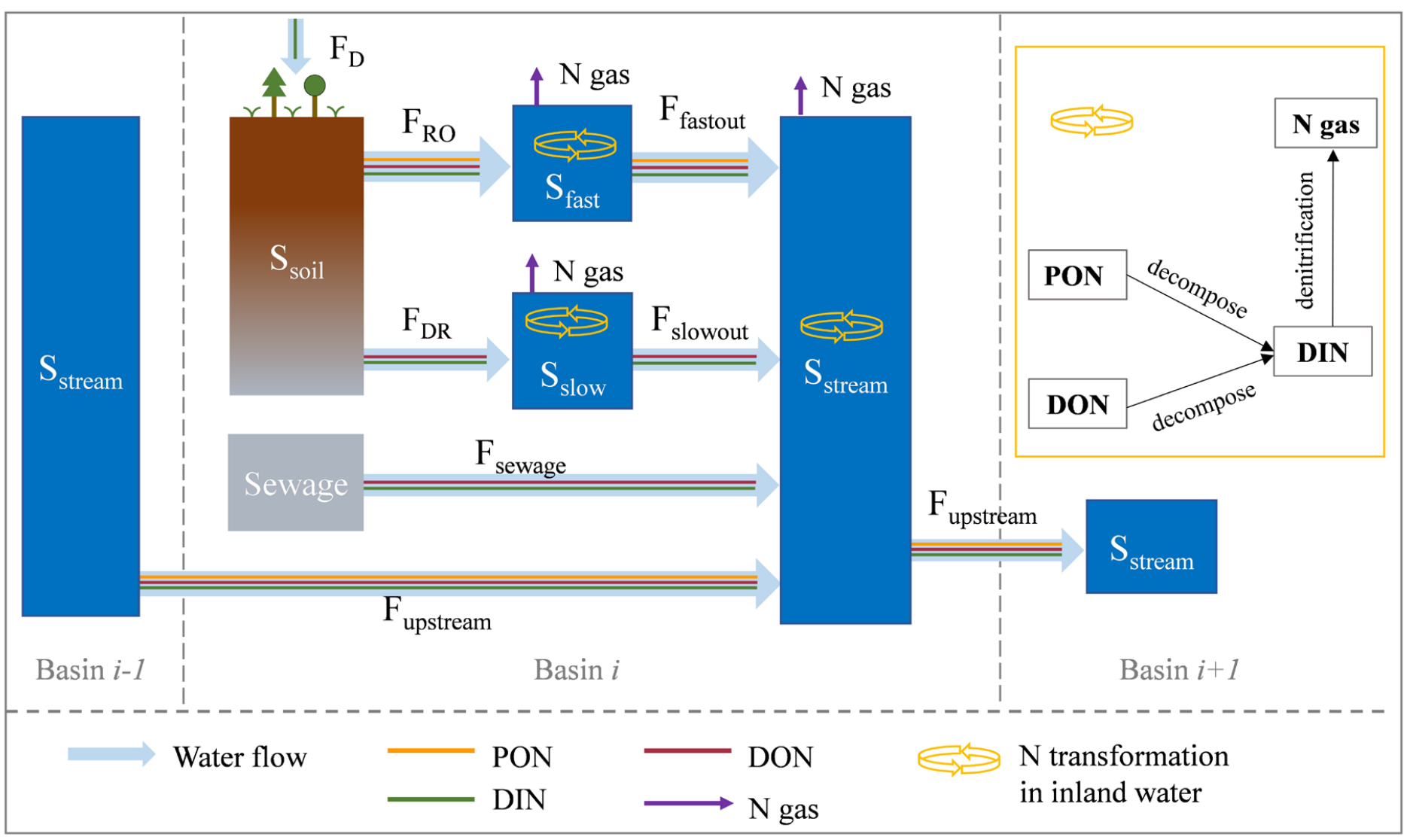

Figure 2. Schematic plot for the reservoirs and flows of water and N in ORCHIDEE-NLAT. $S_{soil}$ is the soil pool. $S_{fast}$, $S_{slow}$, $S_{stream}$ are the "fast", "slow" and stream reservoir, respectively. $F_{RO}$ and $F_{DR}$ are surface runoff and belowground drainage, respectively. $F_{fastout}$ is the flow from fast reservoir to stream reservoir. $F_{slowout}$ is the flow from slow reservoir to stream reservoir.

$F_{upstream}$ and $F_{downstream}$ are the upstream inputs from basin *i-1* and downstream outputs to basin *i+1*, respectively. $F_D$ is the wet and dry deposition of DIN from the atmosphere.

Following the routing scheme of water in ORCHIDEE-NLAT, N contained in surface runoff ($F_{RO}$) and belowground drainage ($F_{DR}$) flows into the fast and slow reservoir, respectively. Subsequently, and depending on the water residence time, the N stocks in these reservoirs are subject to decomposition and losses via denitrification. The remaining fractions flow further into the stream reservoirs, which also receive N inputs delivered directly by sewage (Fig. 2). Within stream reservoirs, N is transformed by biogeochemical reactions and flows from grid cell to grid cell along the river routing scheme. The timescale for biogeochemical transformation processes scale to the water residence time (and hence topography) within the river network, and the fraction of N that is not lost to the atmosphere via denitrification is ultimately exported to the coast. The biogeochemical reactions within each reservoir include the decomposition of PON and DON to DIN, and the denitrification of DIN to N gas which is assumed all released to the atmosphere (Fig. 2). The mass balance equations for the N stocks in different reservoirs are calculated as follows:

$$\frac{dS_{fast_PON}}{dt} = F_{RO_PON} - F_{fastout_PON} - R_{fast_PON} \quad (4)$$

$$\frac{dS_{fast_DON}}{dt} = F_{RO_DON} - F_{fastout_DON} - R_{fast_DON} \quad (5)$$

$$\frac{dS_{fast_DIN}}{dt} = F_{RO_DIN} - F_{fastout_DIN} - R_{fast_DIN} \quad (6)$$

$$\frac{dS_{slow_DON}}{dt} = F_{DR_DON} - F_{slowout_DON} - R_{slow_DON} \quad (7)$$

$$\frac{dS_{slow_DIN}}{dt} = F_{DR_DIN} - F_{slowout_DIN} - R_{slow_DIN} \quad (8)$$

$$\frac{dS_{stream_PON}}{dt} = F_{fastout_PON} + F_{upstream_PON} - R_{stream_PON} - F_{downstream_PON} \quad (9)$$

$$\frac{dS_{stream_DON}}{dt} = F_{fastout_DON} + F_{slowout_DON} + F_{upstream_DON} + F_{sewage_DON} - R_{stream_DON} - R_{downstream_DON} \quad (10)$$

$$\frac{dS_{stream_DIN}}{dt} = F_{fastout_DIN} + F_{slowout_DIN} + F_{upstream_DIN} + F_{sewage_DIN} + R_{stream_PON} + R_{stream_DON} - R_{stream_DIN} - F_{downstream_DIN} \quad (11)$$

where $F_{upstream_PON}$ (g N d$^{-1}$), $F_{upstream_DON}$ (g N d$^{-1}$) and $F_{upstream_DIN}$ (g N d$^{-1}$) represent the inflow rates of PON, DON and DIN, respectively, from upstream grids to the next grid; $F_{downstream_PON}$ (g N d$^{-1}$), $F_{downstream_DON}$ (g N d$^{-1}$) and $F_{downstream_DIN}$ (g N d$^{-1}$) represent outflow rates of PON, DON and DIN from a given grid to downstream grid, respectively. For each N species, the N inputs to a stream reservoir in a given grid cell *i* ($F_{upstream_PON}$, $F_{upstream_DON}$ and $F_{upstream_DON}$ in Eqs. 9-11) is equal to the N outflow from the upstream stream reservoir in the grid cell *i-1* ($F_{streamdown_PON}$, $F_{streamdown_PON}$ and $F_{streamdown_PON}$ in Eqs. 17-19).

We assume that N concentrations are homogeneously distributed within each reservoir of each grid and that the transfers of N from one reservoir to another simply follow that of water. N transfers are calculated according to:

$$F_{fastout_PON} = S_{fast_PON} \times \frac{F_{fastout_H2O}}{S_{fast_H2O}} \quad (12)$$

$$F_{fastout_DON} = S_{fast_DON} \times \frac{F_{fastout_H2O}}{S_{fast_H2O}} \quad (13)$$

$$F_{fastout_DIN} = S_{fast_DIN} \times \frac{F_{fastout_H2O}}{S_{fast_H2O}} \quad (14)$$

$$F_{slowout_DON} = S_{slow_DON} \times \frac{F_{slowout_H2O}}{S_{slow_H2O}} \quad (15)$$

$$F_{slowout_DIN} = S_{slow_DIN} \times \frac{F_{slowout_H2O}}{S_{slow_H2O}} \quad (16)$$

$$F_{streamdown_PON} = S_{stream_PON} \times \frac{F_{streamout_H2O}}{S_{stream_H2O}} \quad (17)$$

$$F_{streamdown_DON} = S_{stream_DON} \times \frac{F_{streamout_H2O}}{S_{stream_H2O}} \quad (18)$$

$$F_{streamdown_DIN} = S_{stream_DIN} \times \frac{F_{streamout_H2O}}{S_{stream_H2O}} \quad (19)$$

where all the $S$ terms represent N (g N) and water stocks ($m^3$), and $F$ terms represent flow rates of water ($m^3$ $d^{-1}$) and N (g N $d^{-1}$) .

Temperature controls the decomposition rates of organic N in rivers (Ferreira et al., 2020). Following the algorithm of Xia et al. (2013), the rates of PON and DON decomposition in each reservoir are calculated using first-order kinetics of the corresponding N stock and a Q10 temperature dependence based on water temperature.

$$R_{fast_PON} = S_{fast_PON} \times K_{PON} \times Q10^{\frac{TW-T_{ref1}}{10}} \quad (20)$$

$$R_{stream_PON} = S_{stream_PON} \times K_{PON} \times Q10^{\frac{TW-T_{ref1}}{10}} \quad (21)$$

$$R_{fast_DON} = S_{fast_DON} \times K_{DON} \times Q10^{\frac{TW-T_{ref1}}{10}} \quad (22)$$

$$R_{slow_DON} = S_{slow_DON} \times K_{DON} \times Q10^{\frac{TW-T_{ref1}}{10}} \quad (23)$$

$$R_{stream_DON} = S_{stream_DON} \times K_{DON} \times Q10^{\frac{TW-T_{ref1}}{10}} \quad (24)$$

$K_{PON}$ (0.028 $d^{-1}$) represents the average PON decomposition rate at 20°C in water (Islam et al., 2012); $K_{DON}$ (0.07 $d^{-1}$) represents the average DON decomposition rate at the reference temperature of 20°C in water (Xia et al., 2013); $Q10$ is the temperature sensitivity of PON and DON decomposition rates (= 2.0 after Liu et al., 2021; Yang et al, 2015); $TW$ is the water temperature (°C); and $T_{ref1}$ is the reference temperature for PON and DON decomposition (=20°C).

The denitrification rates of DIN decrease with stream depth, because most denitrification happens in benthic sediments rather than in the water column, so high benthic area to water volume ratios result in high denitrification rates (Bernot and Dodds, 2005; Aitkenhead-Peterson et al., 2001). In addition, denitrification rates are also controlled by temperature (Jung et al., 2014; Ma et al., 2022). The denitrification is simulated by adapting the equations of Pauer et al. (2009):

$$R_{fast_DIN} = \frac{S_{fast_DIN}}{depth} \times K_{DIN} \times F_{T_DIN} \quad (25)$$

$$R_{slow_DIN} = \frac{S_{slow_DIN}}{depth} \times K_{DIN} \times F_{T_DIN} \quad (26)$$

$$R_{stream_DIN} = \frac{S_{stream_DIN}}{depth} \times K_{DIN} \times F_{T_DIN} \quad (27)$$

$$F_{T_DIN} = e^{\frac{-(TW-T_{ref2})^2}{T_{ref2}{}^2}} \quad (28)$$

$$depth = max\ (e^{2.56} \times Q^{0.423}, 1.0) \quad (29)$$

where $K_{DIN}$ (0.15 $d^{-1}$) represents the denitrification rate in water at 25°C (Alexander et al., 2009); $F_{T_DIN}$ (unitless) represents the dependency of denitrification on temperature (Ma et al., 2022); $T_{ref2}$ is the reference temperature for denitrification (=25°C); Here $\frac{1}{depth}$ (unitless) represents the factor that simulates the role of the benthic surface area to water volume ratio as a key control factor of denitrification rates. The stream *depth* is simulated according to Eq. 29 (Raymond et al., 2012). Therefore, aside from available DIN stocks, denitrification rates are spatially and temporally dependent through the effects of water residence time (controlled by topography), temperature and water depths (controlled by discharge). See Tables A1 and A2 for a summary of all variables, fluxes and processes incorporated in ORCHIDEE-NLAT.

## 2.2. Observational data

Riverine water discharge from the Global Runoff Data Centre (GRDC) (Federal Institute of Hydrology, 2018) and riverine TN and $NO_3^-$ concentrations from the Global River water Quality Archive (GRQA) (Virro et al., 2021) were used to evaluate ORCHIDEE-NLAT (Fig. 3). We retrieved GRDC water discharge data for 350 gauging stations with a catchment area greater than 50 000 km$^2$. From the GRQA data, only time-series with more than two observations in each month of one year were retained for model evaluation. For N concentrations, after removing duplicates in the GRQA database, we collected data of TN for 3507 sites and $NO_3^-$ for 1841 sites. Moreover, as observations of $NO_3^-$ at a given site are generally more frequent and cover a longer time span than for TN, we used the strong correlation between both species to estimate TN concentrations from $NO_3^-$ if only the latter were available (yellow dots in Fig. 3). The prediction equation applied in this study (Eq. 30, Fig. S1) was obtained based on GRQA data at 148 sites with simultaneous concentrations of both TN and $NO_3^-$ ($R^2$=0.78):

$$C_{TN_obs} = 1.33 \times C_{NO3_obs} + 0.56 \tag{30}$$

where $C_{TN_obs}$ (g N m$^{-3}$) and $C_{NO3_obs}$ (g N m$^{-3}$) represent the observed concentrations of TN and $NO_3^-$, respectively.

The TN flow rates equal to the water discharge rates multiplied by N concentrations. Therefore, for a given GRDC site, we systematically selected the nearest GRQA site with reported N concentration (McDowell et al., 2021) to calculate the flux:

$$F_{TN_obs} = F_{W_obs} \times C_{TN_obs} \tag{31}$$

where $F_{TN_obs}$ (g N d$^{-1}$) and $F_{W_obs}$ (m$^3$ d$^{-1}$) represent observed rates of TN flow and water discharge, respectively.

Since TN concentrations for several large rivers (e.g., Amazon and Chinese rivers) were missing in GRQA, we complemented this dataset by

collecting additional observational TN data from peer-reviewed literature (green dots in Fig. 3), resulting in the addition of 20 sites to our database, see details in Table S1.

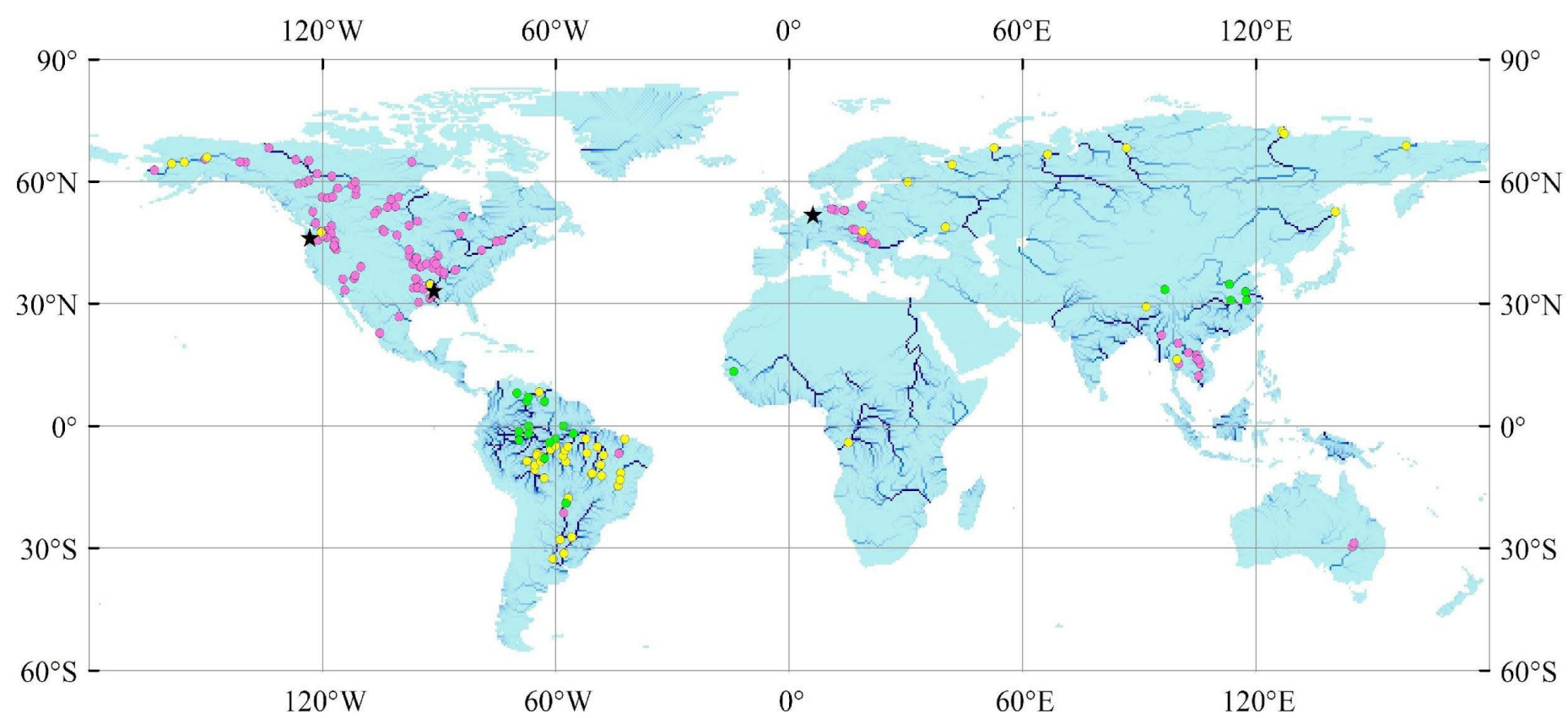

Figure 3. Location of observational sites for N concentrations. Pink dots represent sites with observations of total nitrogen (TN),116 sites; yellow dots represent sites with observations of $NO_3^-$, 53 sites; green dots represent sites with observations of TN from published literature, 20 sites (Table S1); and black stars represent sites with time series of water discharge and TN flow.

## 2.3. Simulation protocol and analysis of model results

### 2.3.1. Simulation protocol

ORCHIDEE-NLAT was applied to simulate the lateral transfer of PON, DON and DIN, as well as the decomposition of PON and DON, and the denitrification of DIN within the river network over the period 1901-2014. The model was run at 0.5° spatial resolution and daily temporal resolution, using the downscaled terrestrial forcings as inputs (see section 2.1.2). Running ORCHIDEE-NLAT at a daily step enables us to evaluate the model performance in simulating not only long-term trends but also the seasonality in lateral N transfers and transformations within the global river network. Model evaluation was conducted at a daily time-step by comparing the amount of simulated and observed TN lateral transfer at three sites with a long time series

of observations for TN flow. We also evaluated the performance of ORCHIDEE-NLAT in simulating annual lateral TN transfer against observational data from the selected 189 sites around the world. The simulated total amounts of PON, DON and DIN from land to river and from river to ocean were further compared with previously published global N models, namely IMAGE-GNM (Vilmin et al., 2018), FrAMES-N (Frame-work for Aquatic Modeling in the Earth System) (Wollheim et al., 2008), MBM (Mass Balance Model) (Green et al., 2004), and Global NEWS2 (Mayorga et al., 2010).

Table 1 summarises the forcing and evaluation data along with their spatiotemporal resolution and references to the gridded products and point datasets.

Table 1. List of forcing data needed to run ORCHIDEE-NLAT and the data used to evaluate the simulation results. $S_{res}$ and $T_{res}$ are the original spatial and temporal resolution of the forcing data, respectively.

| . | Data | $S_{res}$ | $T_{res}$ | Data source |
|---|---|---|---|---|
| Forcing data | Runoff<br>Drainage<br>DOC and POC with runoff<br>DOC and POC with drainage<br>Soil temperature (TS) | 1° | daily | ORCHIDEE-Clateral (Zhang et al., 2022) |
| | DIN with runoff and drainage<br>DON leaching from manure application | 2° | daily | ORCHIDEE-CNP (Sun et al., 2021) |
| | DIN and DON with sewage | 0.5° | yearly | Beusen et al., 2014 |
| | Flow direction<br>Topographic index ($f_{topo}$) | 0.5° | / | Vörösmarty et al., 2000 |
| Evaluation data | Riverine water discharge | / | daily | GRDC[a] |
| | Riverine TN and $NO_3^-$ concentration | / | point measurement | GRQA[b] |
| | Riverine TN concentration | / | point measurement | Table S1 |

[a] Global Runoff Data Centre (GRDC) (Federal Institute of Hydrology, 2018); [b] Global River water Quality Archive (GRQA) (Virro et al., 2021).

### 2.3.2. Model evaluation metrics

To evaluate the performance of ORCHIDEE-NLAT in reproducing the spatial variations of water and N flow, the relative predictive error (RPE) and the coefficient of determination $R^2$ were determined. The $R^2$ represents how much variation in the observations can be explained by the model. The RPE quantifies the extent to which ORCHIDEE-NLAT overestimates or underestimates observations of water discharge and TN flow at grid level.

$$RPE = \frac{M-O}{O} \times 100\% \tag{32}$$

where $M$ is the mean of simulated values, $O$ is the mean of observed values.

To assess the performance of ORCHIDEE-NLAT in reproducing time series of TN and water flows, the relative root mean square root (RRMSE) and Nash-Sutcliffe coefficient (NSE) were determined.

$$RRMSE = \frac{\sqrt{\frac{\sum_{j=1}^{n}(M_j - O_j)^2}{n}}}{\bar{O}} \times 100\% \tag{33}$$

$$NSE = 1 - \frac{\sum_{j=1}^{n}(O_j - M_j)^2}{\sum_{j=1}^{n}(O_j - \bar{O})^2} \tag{34}$$

where $n$ represents the total number of days when observations are available at a given site; $O_j$ and $M_j$ represent observed and modelled values of water/TN flow on day $j$ . NSE can take values between 1 and $-\infty$. An NSE = 1 would mean a perfect fit between observed and simulated values, NSE = 0 means that using the mean observed value as constant simulated value would lead to as much deviation between observed and predicted values as using the actual simulated values. If NSE is negative, there is more deviation between simulated and observed values than between the observed values and their mean.

### 2.3.3. Seasonality analysis

To explore the seasonal variability of water discharge, TN flow, TN concentration and denitrification rates during 2001-2014 at the global-scale, we constructed spatial maps of monthly anomalies following the method by Roobaert et al (2019). The $FV$ represents the relevant flux, rate or concentration, we have that for each grid cell, the monthly anomaly of FV can be calculated as the difference between the FV value for a given month and its annual mean:

$$FVA_t' = FV_t - \overline{FV} \tag{35}$$

where $FVA'_t$ (g N yr$^{-1}$) represent the anomaly of FV in month *t*, while $FV_t$ (g N yr$^{-1}$) and $\overline{FV}$ (g N yr$^{-1}$) represent the values of FV in month *t* and for the annual mean, respectively.

The seasonality, that is the amplitude in seasonal water discharge, N flow rates, N concentrations and denitrification rates, is expressed as the root-mean-square (RMS) of the monthly $FVA'$.

$$season_{VA} = \sqrt{\frac{1}{12} \times \sum_1^{12} (FVA'_t)^2} \quad (36)$$

## 3. Results and discussion

### 3.1. Model evaluation

Evaluation of the water discharge results using the GRDC data indicates that for major rivers with drainage areas larger than 50 000 km$^2$ spread over the globe, ORCHIDEE-NLAT reproduces the magnitude and seasonal variations of water discharge well. Overall, the model simulation explains 90% of the spatial variation in the observed long-term average water discharges (Fig. 4a, Fig. S2 a). The absolute values of RPE for the simulated average water discharges are mostly smaller than 50% (Fig. S3a). At 25 sites (13% of all sites), the absolute values of RPE are larger than 100%, but the annual mean values of water discharge at each of these sites are less than 1.0×10$^{11}$ m$^3$ yr$^{-1}$, indicating that large errors only occur at some sites draining relatively small basins (Fig. S3a). The discrepancy between model and observations at these sites may be caused by two factors: (1) a potentially substantial discrepancy between the stream routing scheme (delineation of catchment boundaries) defined by the forcing data at 0.5° resolution and the real river network; and (2) the presence of stream channel bifurcations that are poorly resolved by the model (Zhang et al., 2022). At some sites, such as the Columbia, Rhine and Mississippi Rivers,

ORCHIDEE-NLAT also captures the seasonal variation of the water discharges well, with RRMSE ranging from 30% to 41% (Fig. 5 a1-a3).

Evaluation of area-averaged TN flows are overall comparable to observed TN flows at the 189 sites extracted from the GRQA database and additional published literature. ORCHIDEE-NLAT explains 77% of the observed spatial variation of long-term TN flows across sites (Fig. 4b, Fig. S2b). The absolute values of RPE of the simulated average TN flows are mostly smaller than 50% (Fig. S3 b). ORCHIDEE-NLAT significantly underestimated (RPE < -100%) or overestimated (RPE > 100%) the observed TN flows at 32 sites (17% of all sites). Similar to water discharge, these sites are all located in relatively small basins with annual water discharge less than $1.0\times10^{11}$ $m^3$ $yr^{-1}$ (Fig. S3 b). At 9 sites (28% of the 32 sites), the RPE of TN flow is very close to that of water discharge, showing that at these sites, the water discharge (and not the N concentrations) is the main reason for the discrepancies between observed and modelled TN flows. The results reveal that the RPE of TN flow is relatively small for large rivers, such as at sites located in the lower reaches of the Columbia, Rhine and Mississippi Rivers, where RPE values are -25%, -16% and 26%, respectively. ORCHIDEE-NLAT also reproduces well the seasonal patterns of TN flow in these rivers, with RRMSE ranging from 30% to 64% (Fig.5 b1-b3). At the Rhine river site, the NSE of TN flow is negative, reveals that although the seasonal pattern of TN flow simulated by ORCHIDEE-NLAT is similar to that observed, it does not capture accurate trends on the day scale (Fig.5 b2).

The simulated DIN concentrations display broadly similar spatial patterns and concentration ranges as obtained from a recent observation based machine-learning (ML) based assessment (Marzadri et al., 2021). ML involves a fair amount of empirical modelling, and this comparison can thus not be considered as a direct model evaluation by data. Nevertheless, the agreement between both

assessments (Fig. S4) lends further confidence in the capacity of our model to realistically simulate the N cycle along the global river network.

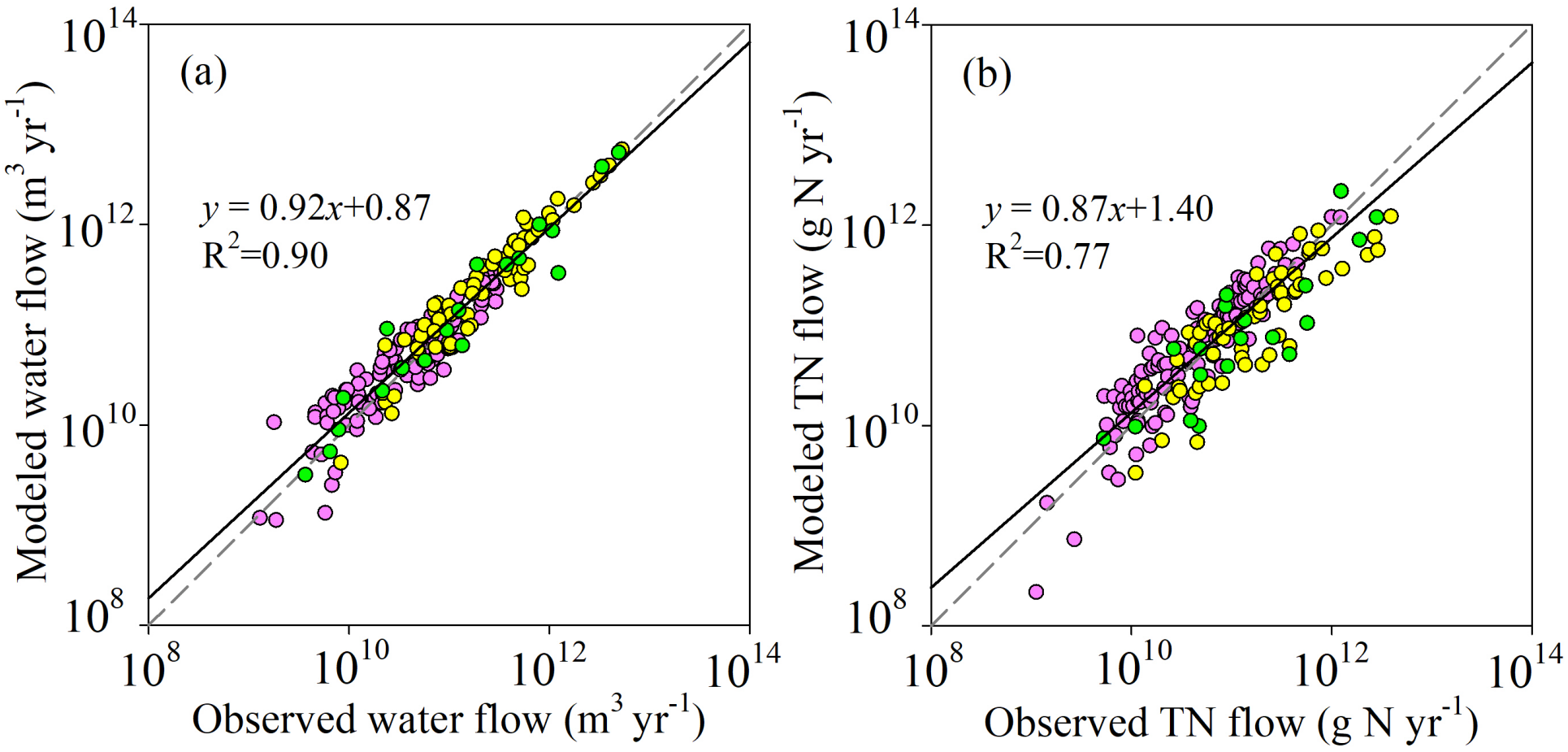

Figure 4. Evaluation of ORCHIDEE-NLAT. Global-scale comparison between observed and modelled annual-mean water discharge (a) and TN flow (b). Pink symbols represent sites with observations of TN from GRQA, yellow symbols represent GRQA sites for which TN concentrations were estimated from observations of $NO_3^-$, and green symbols represent sites with observations of TN from published literature.

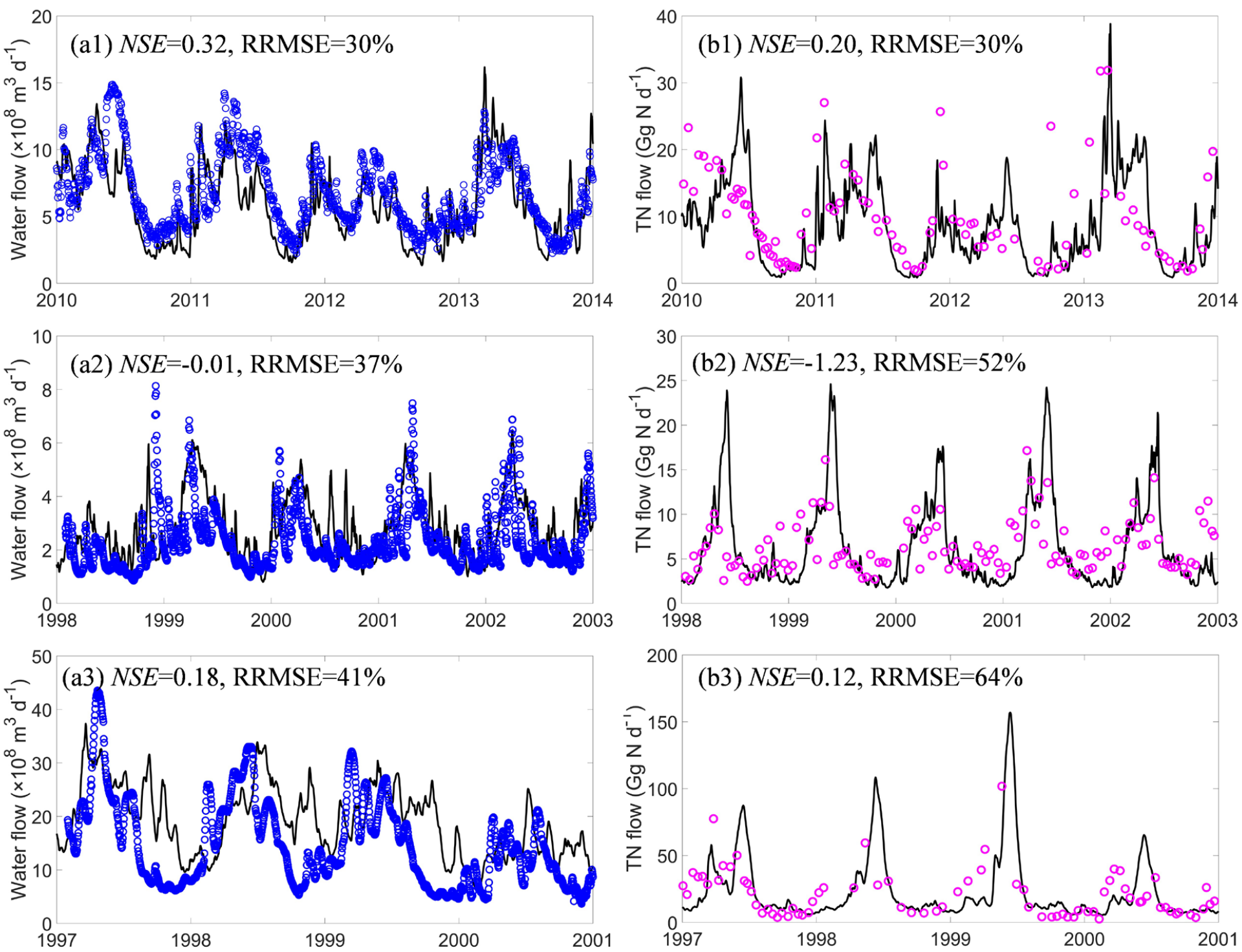

Figure 5. Time series of water discharge (a) and TN flow (b). (a1) and (b1) Columbia-river (46.18°N, 123.18°W); (a2) and (b2) Rhine-river, (51.84°N, 6.11°E); (a3) and (b3) Mississippi river (32.25°N, -91.25°W).

## 3.2. Temporal and spatial patterns of N flows

### 3.2.1. Trends in global N flows

Averaged over the 2001-2014 period, the annual TN input from soils to rivers, TN exports to oceans and denitrification in transit amount to 67.4 Tg N $yr^{-1}$, 40.8 Tg N $yr^{-1}$, and 26.6 Tg N $yr^{-1}$, respectively. These three N fluxes show increasing trends from 1901 to 2014. The global annual TN input to rivers increased by 82.3 %, from 36.8 Tg N $yr^{-1}$ during 1901-1910 to 67.4 Tg N $yr^{-1}$ during 2001-2014 (Fig. 6 a). The global annual TN exports to oceans increased by 50.4 % from 27.1 Tg N $yr^{-1}$ to 40.8 Tg N $yr^{-1}$. Most of the increase in N exports to oceans is from DIN which doubled over the simulation period, from 9.6 Tg N $yr^{-1}$ to 20.8 Tg N $yr^{-1}$, while DON exports show a much smaller but

still substantial increase of 56.9% (Fig. 6b). In contrast, PON exports to oceans show a slightly decreasing trend. The increase in global denitrification mostly follows that of increasing DIN inputs, with a relative increase of 174.0 %, from 9.7 Tg N $yr^{-1}$ to 26.6 Tg N $yr^{-1}$. The global TN input into rivers, TN exports to oceans and the denitrification in rivers all show a small peak between 1926 and 1931 due to the relatively higher surface runoff but lower belowground drainage during this period (Fig. S5). The reality of this transient peak is however questionable as it results mostly from meteorological forcing, which is uncertain for the beginning of the $20^{th}$ century.

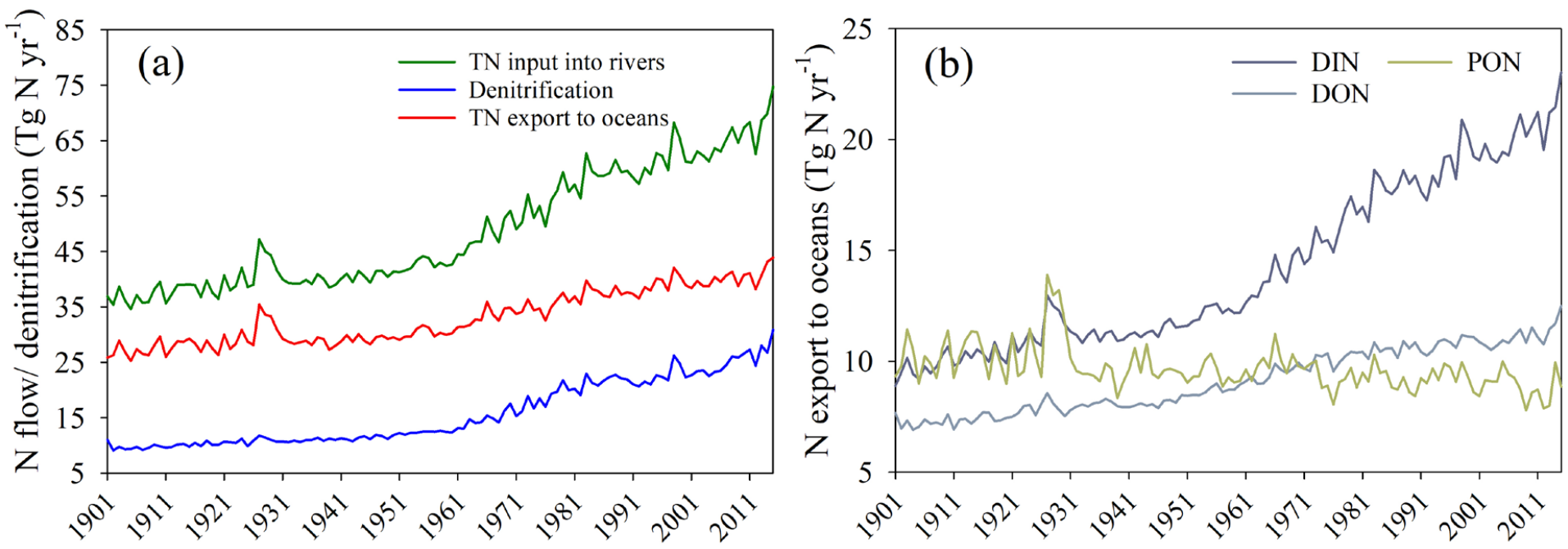

Figure 6. Trends in global N flows from 1901 to 2014: (a) TN inputs into rivers, TN exports to oceans and denitrification; (b) DIN, DON and PON exports to oceans. TN: total nitrogen; DIN: dissolved inorganic nitrogen; DON: dissolved organic nitrogen; PON: particulate organic nitrogen.

### 3.2.2. Spatial patterns in N flows and concentrations

Annual mean TN input into rivers during 2000-2014 shows large spatial heterogeneity, with higher values mainly located in eastern North America, South America, Western Europe, tropical Africa, South Asia, Southeast Asia and Southeast China (Fig. 7a). When compared with 1901-1910, the TN inflow into rivers increased in most areas (about 70%), with the highest increase in China exceeding 300% (Fig. 8a). Annual mean contemporary denitrification rates (2001-2014) also reveal large spatial heterogeneity (Fig. 7b) with high denitrification rates in large tropical and subtropical rivers, for example, the

Amazon river, the Nile river and the Congo river. Over the entire simulation period, the grids with highest relative denitrification increases are mostly located in the subtropics (Fig. 8b).

The 2001-2014 TN export to oceans also varies substantially across regions (Fig. 7c). The riverine TN exports are relatively low for the Arctic Ocean, the western and southern coasts of Australia, and the coastal zone adjacent to desert areas in South America (e.g., the Atacama Desert and the Patagonian Desert), Africa (the Sahara Desert and the Namib Desert), and Asia (e.g., the Arabian Desert, the Thar Desert in India, the deserts of Eastern Iran, and the Syrian Desert) (Fig. 7c). On the contrary, the Amazon region in South America, African rainforest region, Western Europe, South Asia, and southeast China are prominent hot spots of riverine TN exports (Fig. 7c). Unsurprisingly, the TN export to oceans increased in most regions since the beginning of the $20^{th}$ century (Fig. 8c) and in regions such as the south-eastern coastal areas of China, not only the recent TN exports to oceans are relatively high, but also the percentage increase over the $20^{th}$ century exceeded 100% (Fig. 7c and Fig. 8c).

Annual mean contemporary concentration of TN at river mouths also shows large spatial heterogeneity (Fig. 7d), which differs from that of TN export to oceans (Fig. 7c). For instance, the Amazon region is one of the hotspots for TN exports, but its TN concentrations are low (<1 gN $m^{-3}$), because the water discharge and denitrification rates are both high (Fig. 7b, Fig. S6 a). The highest TN concentrations (>5 gN $m^{-3}$) are found in areas with intense human activity, for example San Francisco area, Peru, Spain, Egypt (Nile River estuary) and southeastern coastal areas of China (Bu et al., 2019; Hou et al., 2022; Yang et al., 2023).

The spatial distribution of changes in TN concentrations from 1901-1910 to 2001-2014 is also distinct from that of TN exports (Fig. 8c, d). For example, along the eastern coast of Amapá state in Brazil, and the western coast of
Guinea, Sierra Leone, and Libya, TN exports to the oceans decreased by more than 20%, but TN concentrations increased by more than 10% (Fig. 8c, d). This phenomenon is due to negative trends in water discharge from the corresponding watersheds (Fig. 9, Fig. S6). In most regions, the ratio of TN concentration changes to TN flux changes is between 0 and 1, meaning that TN flux changes are the result of the joint action of changes in water and TN concentrations (TN inputs into rivers) (Fig. 9).

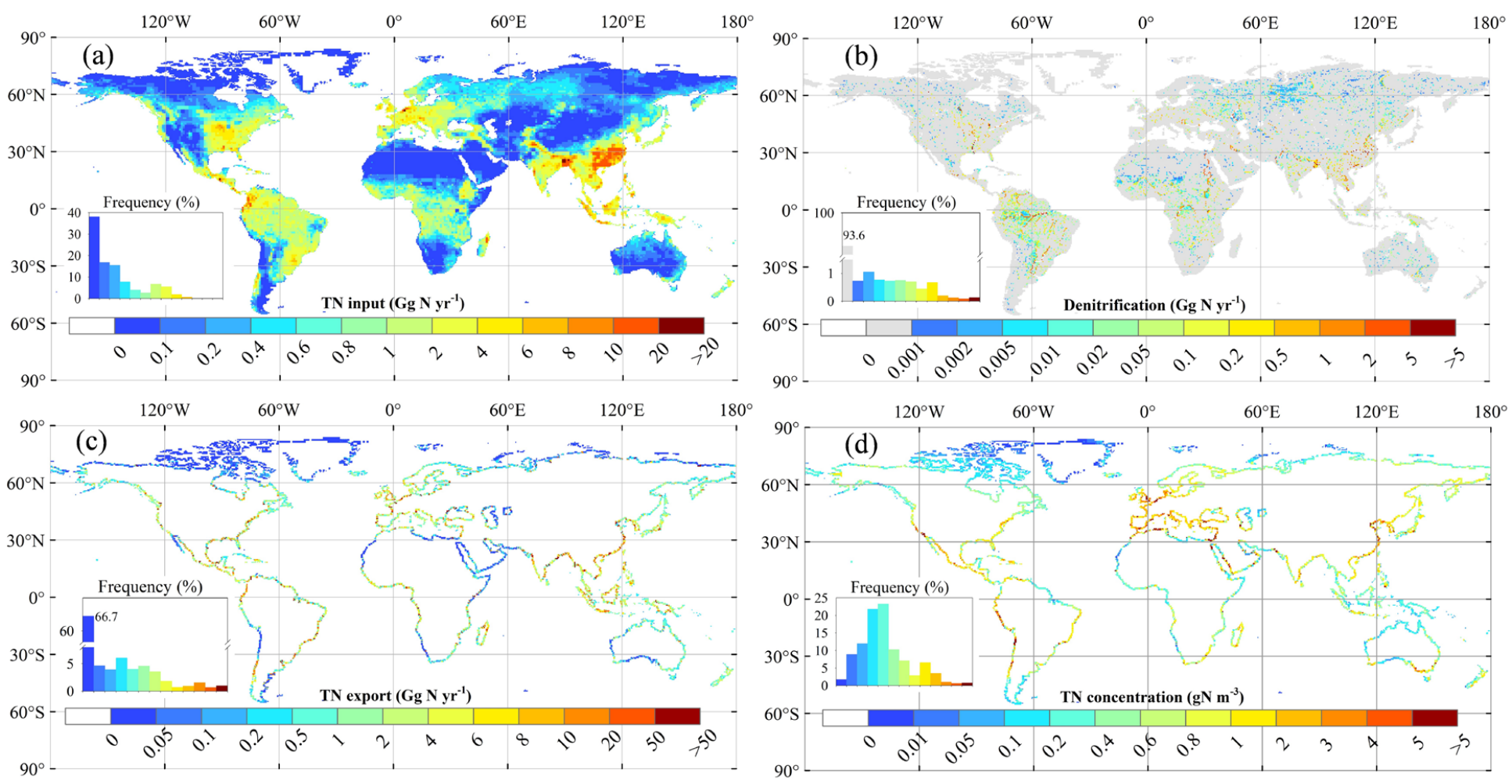

Figure 7. Spatial patterns of annual mean N fluxes and concentrations during 2001-2014: (a) TN inputs into rivers; (b) denitrification rates in rivers; (c) TN exports to oceans; (d) TN concentrations at rivers mouths. To display the spatial patterns of denitrification in rivers better, we excluded data with denitrification rates less than 0.001 GN $yr^{-1}$ per grid.

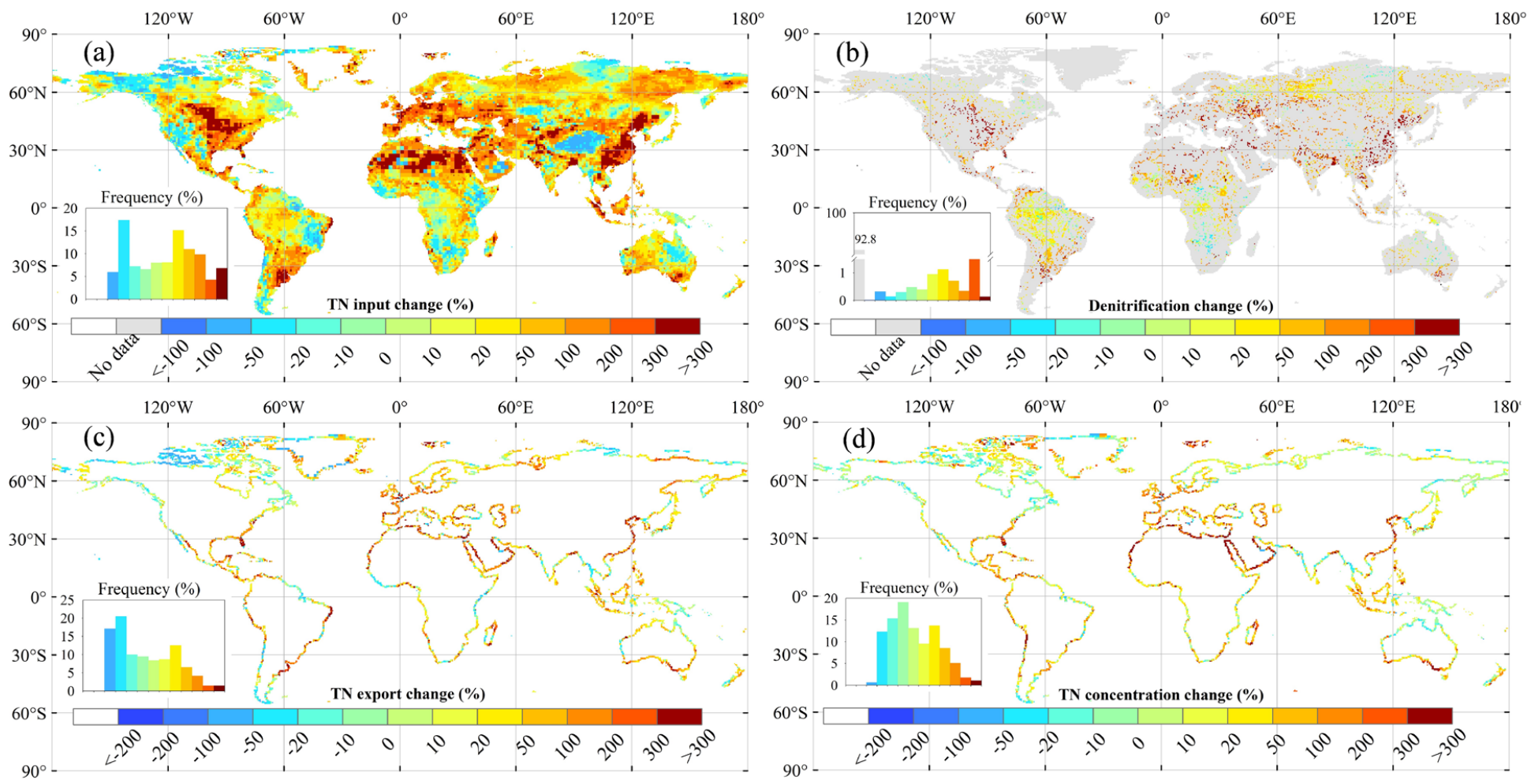

Figure 8. Spatial patterns of changes from 1901–1910 to 2001–2014 of: (a) TN inputs into rivers; (b) denitrification; (c) TN exports to oceans; (d) TN concentrations.

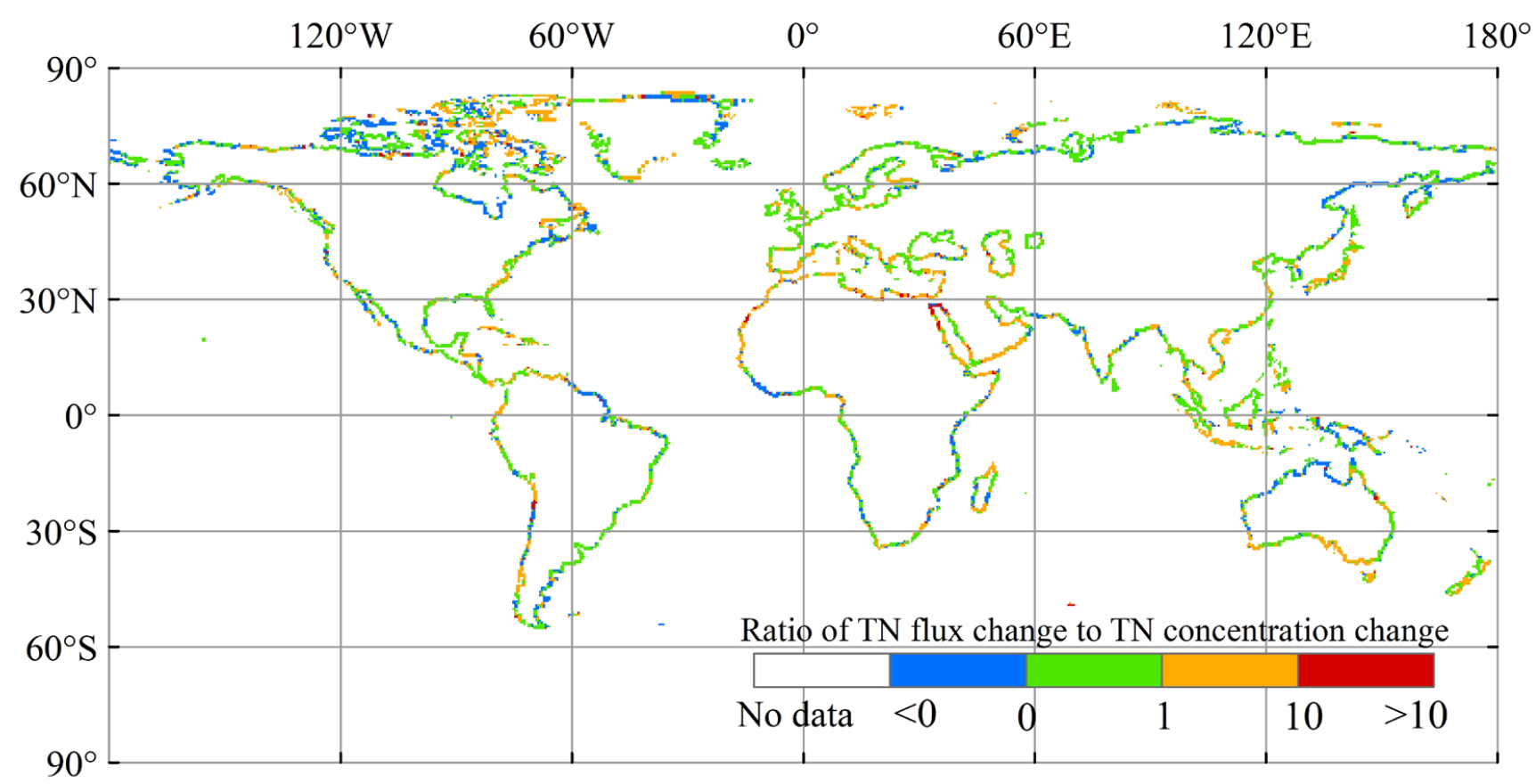

Figure 9. Ratio of TN exports changes to TN concentration changes from period 1901–1910 to 2001–2014.

### 3.2.3. Seasonal variability in N flows and concentrations

The seasonality of TN inputs into rivers over 2001-2014 is most pronounced in the central United States, Europe, South Asia, Southeast Asia and southeast China (Fig. 10a). The frequency distribution of the seasonal amplitude in inputs (Fig.10a) is broadly similar to that of the mean annual inputs (Fig 7a), suggesting a seasonal variability of similar magnitude than the

broad, global scale spatial variability. A similar finding can also be observed for the denitrification rates, with seasonal and spatial variations of the same order of magnitude for both (Fig. 7b, 10b).

The seasonal amplitudes of TN exports to oceans over 2001-2014 shows highest values (> 10 Gg N $yr^{-1}$) along South Asia, and southeast China, and to a lesser extent (1-10 Gg N $yr^{-1}$) along the coastline of the Amazon region, the rainforest regions of Africa, Western Europe, and Mexico (Fig. 10c). Unsurprisingly, a large share of this seasonal variability is due to the river discharge (Fig. S7 a). Our results suggest that the seasonality of TN concentrations at the rivers' mouths has different spatial pattern with seasonal amplitudes of TN exports (Fig. 10c, d). This result is important because the ocean biogeochemical modelling community typically uses annual mean TN fluxes derived from Global News to force their simulations, an downscale these inputs to monthly values under the assumption that the seasonal variability of the flux is entirely due to the river discharge. Our simulations thus stresses the need for models explicitly resolving the seasonal variability of fluxes and concentrations.

We also normalized seasonalities by the mean value of nitrogen flux or concentrations. For TN inputs into reivers, denitrification and TN exports, normalized seasonal maps all show higher values in the middle and high latitudes of the Northern Hemisphere and lower values in the low latitudes and the Southern Hemisphere (Fig. S8). And it is obvious that the regional heterogeneity of normalized seasonality of TN concentration is much weaker than that of nitrogen flux (Fig. S8).

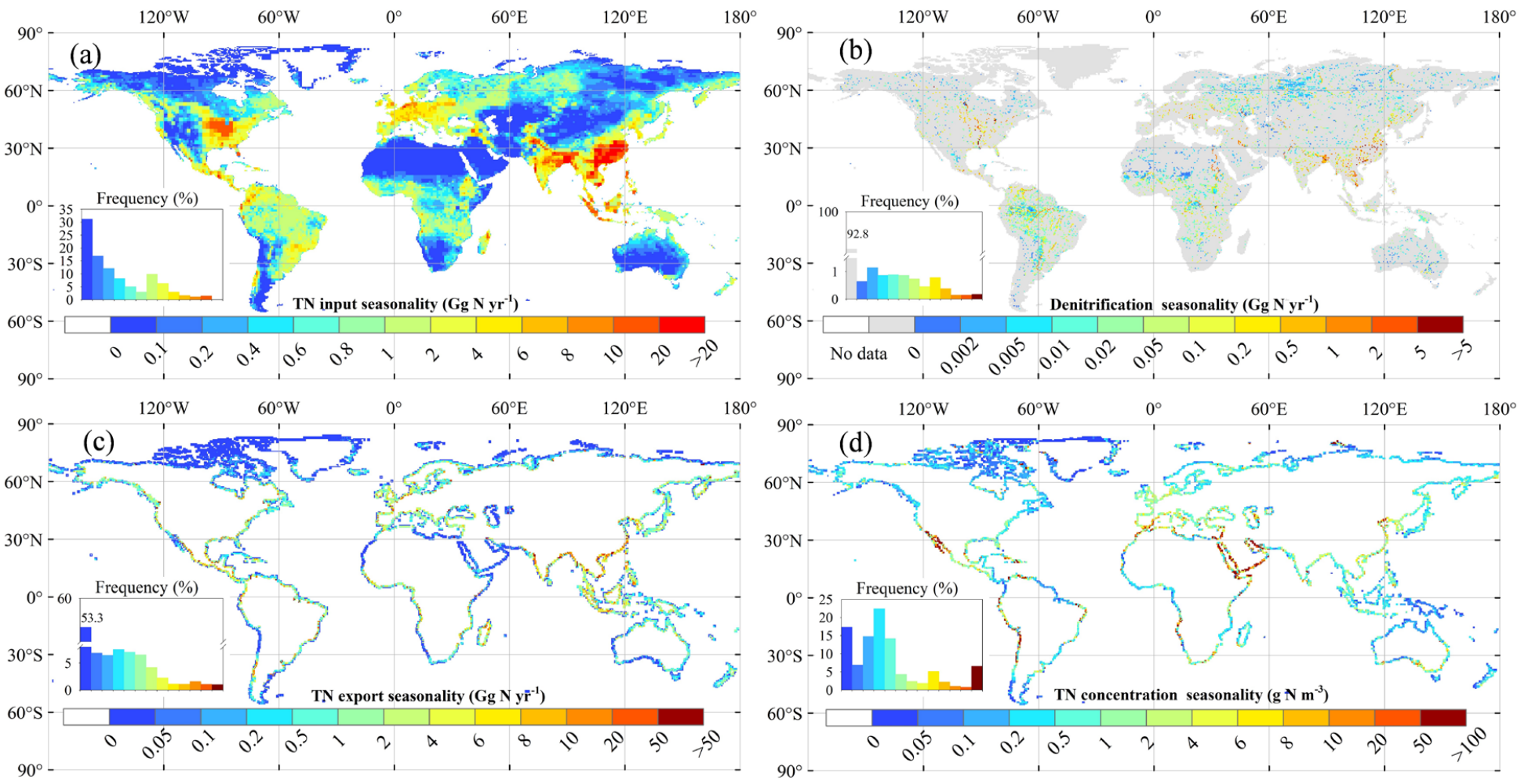

Figure 10. Spatial distribution of seasonality for TN and denitrification over 2001-2014: (a) TN inputs into rivers; (b) TN exports to oceans; (c) dnitrification removal rates; (d)TN concentrations at rivers mouths.

### 3.3. Comparison with other models

We compared the trends of global N input to rivers simulated by ORCHIDEE-NLAT and the recently published IMAGE-GNM (Vilmin et al., 2018). Overall, both models capture a similar increasing trend of global DIN delivery from land to rivers from 1901 till 2001 (Fig. 11a). During 1961-2000, the global-scale interannual variability of DIN simulated by ORCHIDEE-NLAT is comparatively stronger than that simulated by IMAGE-GNM (Fig. 11a). To some extent, this could be due to the different temporal resolution of the two models (daily for ORCHIDEE-NLAT, yearly for IMAGE-GNM) and their associated climate forcings. That is, ORCHIDEE-NLAT calculates annual means from daily fluxes, while IMAGE-GNM does not resolve the intra-annual variability. The results however markedly differ regarding organic N (ON=PON+DON) with IMAGE-GNM simulating a significant increase from 24.9 Tg N $yr^{-1}$ during 1901-1910 to 37.9 Tg N $yr^{-1}$ in during 1990-2000, while the ON inflow simulated by ORCHIDEE-NLAT shows a weaker increasing trend over the same period (26.5 Tg N $yr^{-1}$ during 1901-1910 to 32.4 Tg N $yr^{-1}$

during 1990-2000). The weaker trend in our model can primarily be explained by the increasing DON inflow being offset by a decreasing PON inflow (Fig. 11c). The fundamental reason for the discrepancy among the two models stems from their distinct structures and algorithms. In ORCHIDEE-NLAT, the ON flows into rivers are calculated separately for the dissolved and particulate compounds using a process-based representation of the soil C stock dynamics and C:N ratios, as well as the rates of runoff and drainage. The approach is different in IMAGE-GNM which calculates the bulk ON flows (DON+PON) based on empirical formulas (Vilmin et al., 2018). Specifically, it calculates the ON delivery from land to rivers with drainage based on the TN delivery rate, assuming that 50% of this flux is in the form of ON. For ON flows into rivers with runoff, IMAGE-GNM distinguishes two runoff mobilisation pathways, i.e. losses from recent nutrient applications in forms of fertiliser and manure, and a memory effect related to long-term historical changes in soil nutrient inventories. These two pathways are simulated based on empirical formulas (Vilmin et al., 2018). In ORCHIDEE-NLAT, we used default C:N ratios (from ORCHIDEE-Clateral) in different SOM pools to calculate the PON flow out of soils, and a constant C:N ratio (averaged values from references) to simulate DON flow out of soils. The assumption of constant C:N ratio for dissolved matter in soil could to some extent contribute to the weaker trend in ON delivery to rivers simulated by ORCHIDEE-NLAT, since some studies have revealed that DOC:DON ratios vary with time and land cover (Li et al., 2019; Yates et al., 2019).

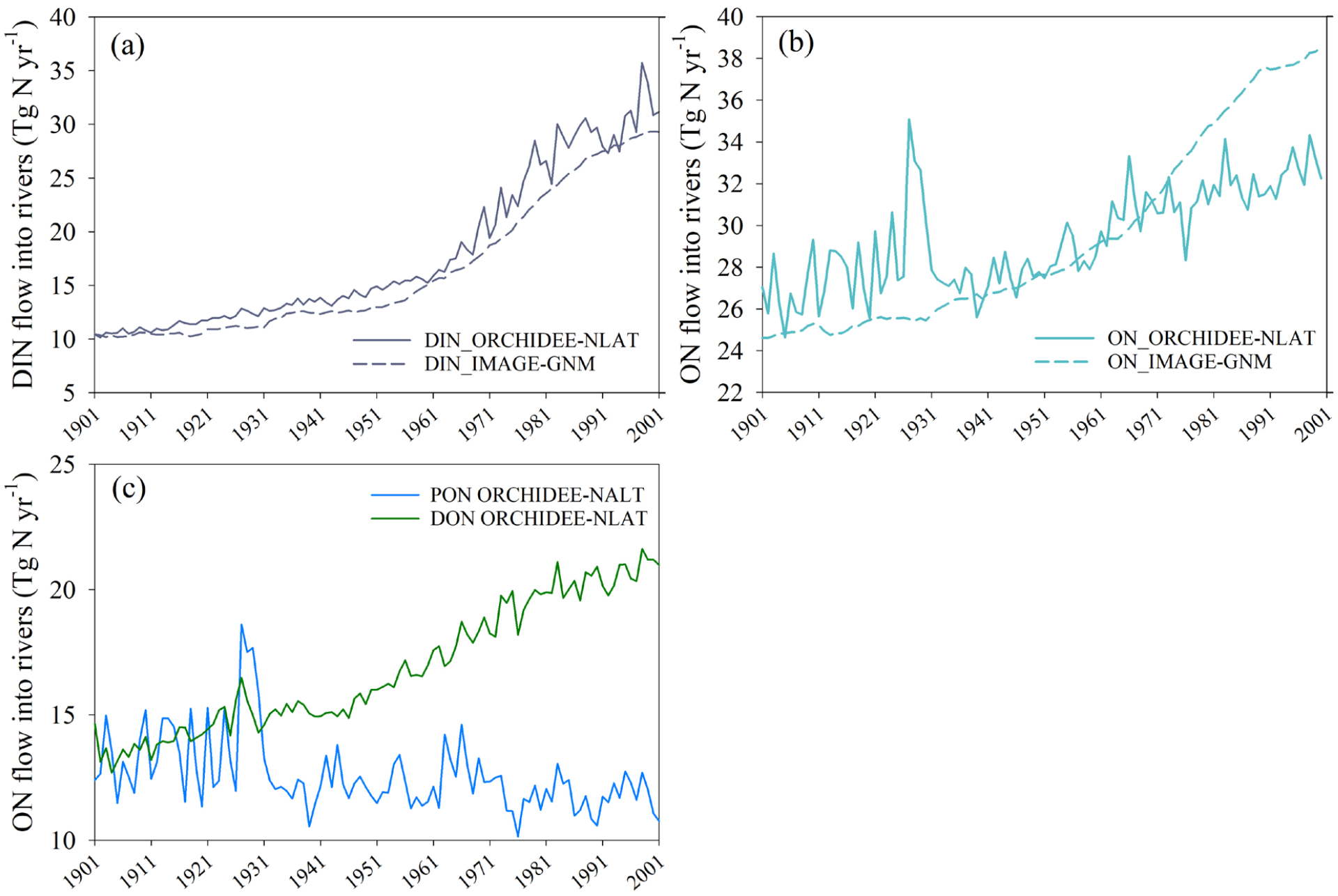

Figure 11. Global terrestrial N flows into rivers from 1901 to 2001 simulated by ORCHIDEE-NLAT and IMAGE-GNM (Vilmin et al., 2018): (a) DIN; (b) ON (DON+PON); (c) DON and PON simulated by ORCHIDEE-NLAT.

The simulated lateral N flows from land to rivers and N exports to oceans in this study are now compared with those simulated by other models for different time horizons, noting that each model covers different time periods (Fig. 12a). Focusing first on the global N flows from land to rivers, we find that for different time horizons, the simulated N input by ORCHIDEE-NLAT is very close with those estimated by IMAGE-GNM (Vilmin et al., 2018) and FrAMES-N (Wollheim et al., 2008) with differences between ORCHIDEE-NLAT and other models for the different time horizons never exceeding 7%. Although the fraction of DIN in TN over 1901-1910 simulated by ORCHIDEE-NLAT (27%) is slightly lower than that of IMAGE-GNM (29%), the DIN fractions simulated by these two models both show obvious increasing trends with time, ORCHIDEE-NLAT and IMAGE-GNM reporting DIN fractions for the 1991-2000 period reaching 48% and 43%, respectively. A comprehensive

cross-biome assessment of N composition in rivers also found that the dissolved N pool shifts from highly heterogeneous to primarily inorganic N in response to human disturbances (Wymore et al., 2021). Changes in the composition of TN inputs from land to rivers is primarily caused by the excess inorganic N release from agricultural (due to the utilisation of fertilisers) and urban (due to the release of sewage) areas.

The global N export from rivers to oceans simulated by ORCHIDEE-NLAT is also comparable to the estimates from other models. During 1901-1910, the global riverine N export to oceans is 29.0 Tg N $yr^{-1}$, within the range of values simulated by IMAGE-GNM (19.0 Tg N yr-1, Vilmin et al., 2018) and DLEM (29.4 Tg N $yr^{-1}$, Tian, pers. com.) (Fig. 12b). For the most recent period (2000s), the simulated riverine N export to oceans is converging, with differences smaller than 10 % compared to other models such as GlobaNEWS2 (Mayorga et al., 2010), IMAGE-GNM, and DLEM (Fig. 12b). Although the global riverine TN export to oceans simulated by ORCHIDEE-NLAT is close to that simulated by GlobalNEWS2 (1970-2010), the TN export reported here contains a slightly larger fraction of DIN and a slightly lower fraction of PON compared to GlobalNEWS2 (Fig. 12b).

The TN export to oceans simulated by ORCHIDEE-NLAT and GlobalNEWS2 are also comparable at continental scale (Fig. 13a), with largest TN exports from Asia, and lowest exports from Australia. However, the simulated proportions of N species in the overall TN export show distinct behaviour between these two models. For example, compared to GlobalNEWS2, the DIN proportion in TN exports simulated by ORCHIDEE-NLAT is larger in Asia, Africa and South America but smaller in Europe (Fig. 13a).

The magnitude of TN exports simulated by ORCHIDEE-NLAT and GlobalNEWS2 continue to diverge at basin scale (Fig. 13b). In 8 of the top 20

basins by area, the difference between the two models is less than 50%, such as the Congo, the Mississippi, the Ob, the Parana, the Yenisei, the Changjiang, the Mackenzie and the Nelson. Larger discrepancies can even be observed for several large river systems. For instance, in the Amazon basin, the TN export simulated by GlobaNEWS2 is about 2.5 times larger than that simulated by ORCHIDEE-NLAT. Evaluation of ORCHIDEE-NLAT simulation results against measurements of TN flow rates in the Amazon River indicates that ORCHIDEE-NLAT underestimates the TN flow in this basin (Fig. S2). At the Manacapuru and the Óbidos, two observation sites on the main channel of the Amazon River, the observed TN flow is 1.90 Tg N $yr^{-1}$ and 2.82 Tg N $yr^{-1}$, but the simulated values are 0.92 Tg N $yr^{-1}$ and 1.57 Tg N $yr^{-1}$, respectively. To evaluate whether the underestimation is caused by less TN inflow into rivers, we set the river transformation processes to zero, and found that the TN flow is 1.56 Tg N $yr^{-1}$ at the Manacapuru site and 2.35 Tg N $yr^{-1}$ at the Óbidos site. Therefore, even with no N removal ORCHIDEE-NLAT still underestimates the observed TN flows at these two sites , a finding suggesting that N delivery from terrestrial ecosystems is too low in the Amazon basin by ORCHIDEE-NLAT. In the Nile basin, the TN export simulated by ORCHIDEE-NLAT is thirty times larger than that simulated by GlobalNEWS2. The observed annual exports of DIN and DON amount to 0.079 Tg N $yr^{-1}$ and 0.038 Tg N $yr^{-1}$, respectively (Badr, 2016). These observed values are of the same magnitude with those of ORCHIDEE-NLAT reaching 0.113 Tg N $yr^{-1}$ for DIN and 0.048Tg N $yr^{-1}$ for DON. suggesting that our model better captures the observed N export for this specific basin than GlobalNEWS2.

It should be noted that the GlobalNEWS2 and IMAGE-GNM both have IMAGE part to simulated N inputs into inland water, but they were developed based on different hydrological models and use different methods to calculate N transport and retention. The hydrological model used in GlobalNEWS2 is Water

Balance Model ($WBM_{plus}$) (Fekete et al., 2010) , and NEWS models were used to calculate nutrient retention in streams and reservoirs (Seitzinger et al., 2005, 2010; Mayorga et al., 2010). The hydrological model ued in IMAGE-GNM is Global Water Balance (PCR-GLOBWB) (Van Beeket al., 2011), and IMAGE-GNM uses the nutrient spiraling approach (Newbold et al., 1981) to describe in-stream retention of both N and P with a yearly time step (following Wollheim et al., 2008).

In summary, the global total N input to rivers and N export to oceans simulated by the different models are comparable, but the spatial distribution of N export to oceans at finer spatial scales shows increasing discrepancies, as does the chemical speciation. This is mainly due to differences in model structures, spatial and temporal resolutions and forcing data. Albeit our model has been evaluated against the largest dataset of river discharge and N concentrations from the recently assembled global GRQA database, the significant cross-model discrepancies that emerge as the analysis is refined to regional patterns and single species urgently calls for ensemble-means assessments, similar to what has recently been performed for C exports to the ocean (Liu et al., 2024).

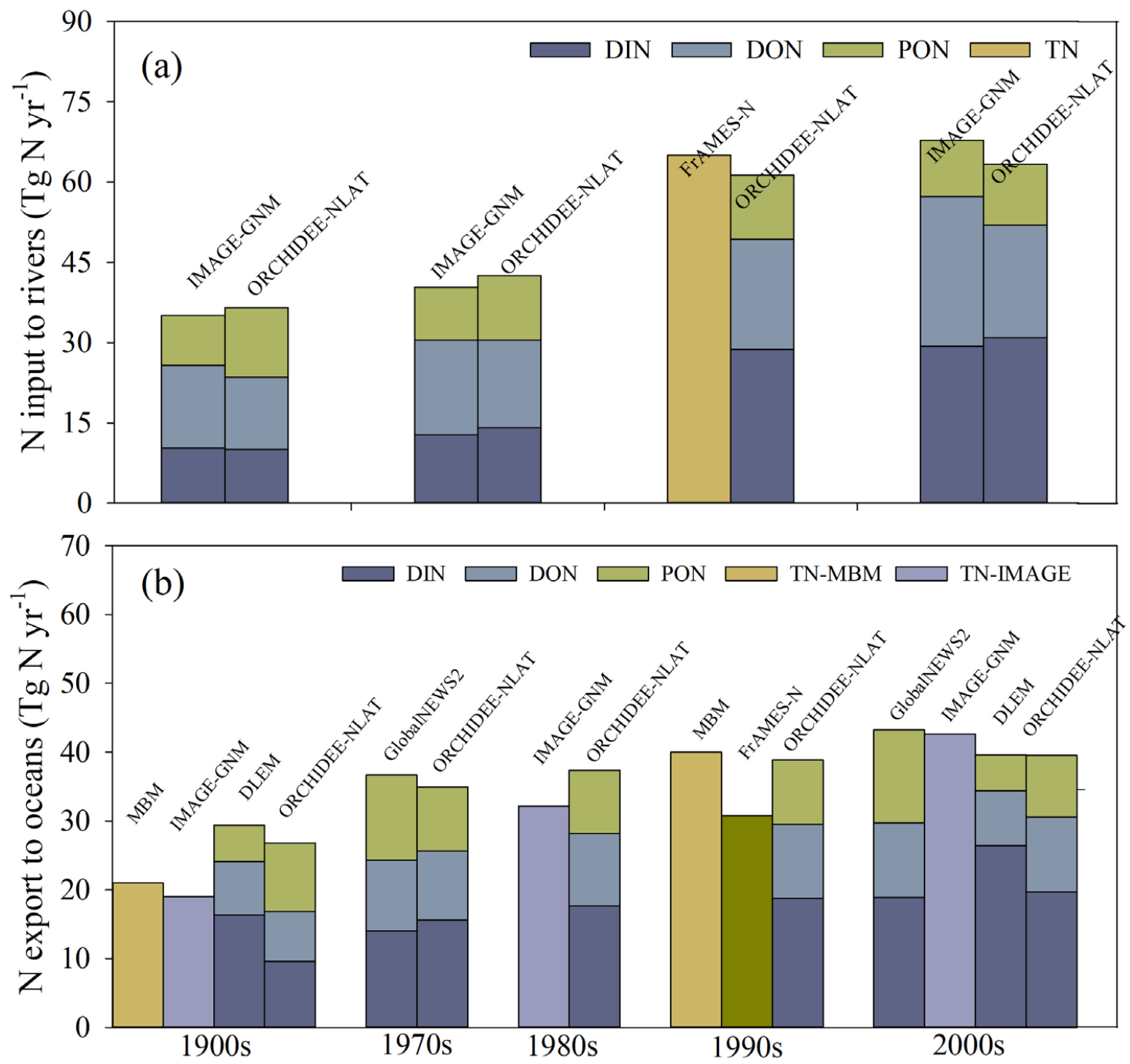

Figure 12. Comparison of global TN fluxes estimated by different models: (a) global TN inputs to rivers; (b) global TN exports to oceans. IMAGE-GNM: Integrated Model to Assess the Global Environment-Global Nutrient Model (Vilmin et al., 2018); FrAMES-N: Frame-work for Aquatic Modeling in the Earth System (Wollheim et al., 2008); MBM: Mass Balance Model (Green et al., 2004); Global NEWS2: Global Nutrient Export from Watersheds 2 (Mayorga et al., 2010); DLEM, Dynamic Land Ecosystem Model, unpublished (Tian, pers. com.).

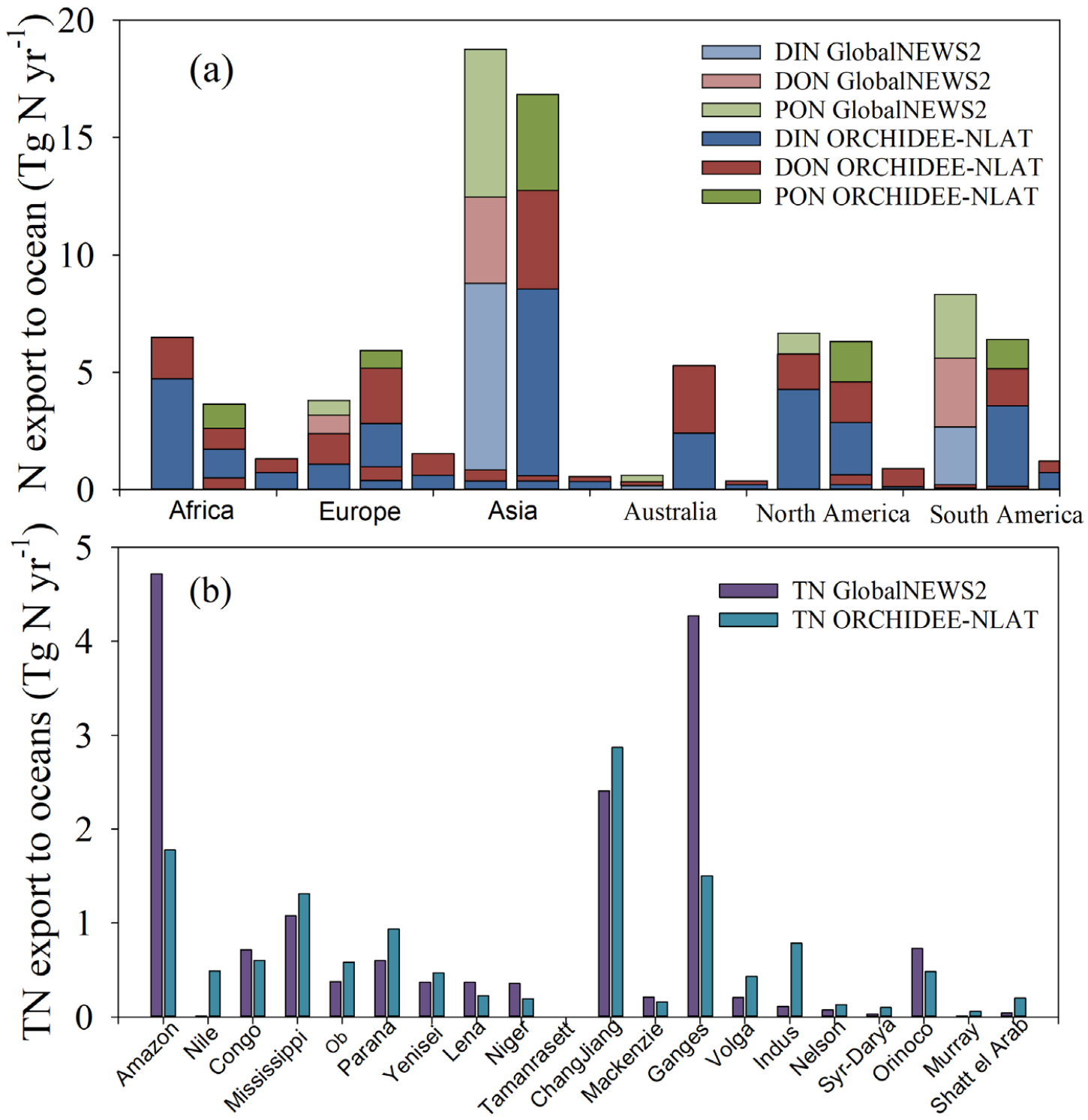

Figure 13. Comparison of TN export to oceans simulated by ORCHIDEE-NLAT and GlobalNEWS2: (a) at continental scale over 2001-2010 (Mayorga et al., 2010); (b) at basin scale over 2001-2010.

## 3.4. Some limitations to keep in mind

ORCHIDEE-NLAT currently relies on a simplified representation of the N processes in benthic sediments and water, without explicit simulation of the hyporheic exchange between sediments and water, instead estimating the importance of these processes via a scaling factor based on water depth. This scaling itself relies on a coarse estimate of the stream channel geometry based on empirical formulas (Raymond et al., 2012). Global-scale databases on the geomorphic properties of global river channels  including river depth and width however exist (Andreadis et al., 2013) and could be used in the future to further refine the representation of N processes in river channels, including the hyporheic exchange between sediments and water. The use of a constant C:N

ratio to simulate DON fluxes from soils to rivers is another limitation of ORCHIDEE-NLAT, since it has been shown to vary over time and land cover type (Li et al., 2019; Yates et al., 2019). In addition, few studies have focused on the influence of PON deposition and resuspension on lateral N transfer in rivers because of the difficulty to represent these processes on the global-scale.

The role of autotrophic production is another process currently omitted Autotrophs (aquatic macrophytes, algae, cyanobacteria, bryophytes, some protists, and bacteria) in freshwater consume N through photosynthesis (King et al., 2014) and may play a significant role in river N cycling. For instance, a long-term study has shown that as pollution from industrial and wastewater emissions decreased, in-situ gross primary production increased, prompting a shift from heterotrophic-dominated (i.e. controlled mainly by nitrification and denitrification) towards autotrophic-dominated N retention regime in rivers (Wachholz et al., 2024). In the future, the role of autotrophic production on N retention should thus be considered, although the large dominance of the heterotrophic metabolism on a global scale suggests that in-situ aquatic production is likely a second-order control (Battin et al., 2023).

In the present version of ORCHIDEE-NLAT, river-floodplain dynamics and channel erosion are currently not represented, because of the incomplete understanding of the effects of these processes on lateral N transfer, the lack of reliable parameters from field studies to calculate the effects of these processes at global scale. Floodplain inundation does not only carry N into river, but also has a significant impact on N retention efficiency in rivers (Martí et al., 1997; Hanrahan et al., 2018), and N cycling (e.g., nitrification and denitrification) in flooded soils (Sánchez-Rodríguez et al., 2019; Hu et al., 2020). For instance, in the Jiulong River watershed, southeast China, flood events exported 47% and 42 % of the annual land-derived ammonium ($NH_4^+$) and $NO_3^-$, respectively, although they only occurred 24% of the time (Gao et al., 2018).

ORCHIDEE-NLAT includes the major sources of riverine N with runoff and drainage in natural, agricultural and urban ecosystems (Fig. 1). Yet, several sources are still missing, for example atmospheric N deposition directly onto rivers and N release from aquaculture (Filoso et al., 2003; Bouwman et al., 2013; Beusen et al., 2016; Gao et al., 2020), suggesting that the N exports to oceans simulated by ORCHIDEE-NLAT might be conservative. On the other hand, N retention and recycling in lakes and artificial reservoirs are currently missing, which have the potential to decrease lateral N flows because they offer ideal conditions for N burial in sediment or permanent loss via denitrification (Saunders & Kalff, 2001; Harrison et al., 2009; Akbarzadeh et al., 2019).

Forcing data used by the ORCHIDEE-NLAT (Table 1) introduces additional uncertainties in the simulation results. The routing scheme of water and N is driven by a map of streamflow direction at 0.5◦ spatial resolution (Vörösmarty et al., 2000, https://doi.org/10.1016/S0022-1694(00)00282-1). There are obvious discrepancies between this routing scheme and the real river network (Zhang et al., 2022). Deviation of flow direction induces uncertainties in the simulated riverine water discharge and N flow because the flow direction directly determines the area of each catchment and the routing of the river.

Finally, although ORCHIDEE-NLAT reproduces the magnitude and seasonal variations of water and N transfer from land to rivers and oceans well (Fig. 4, 5), spatial and temporal biases in observational data also affect the evaluation of model results. Most observations of riverine N are distributed in North America, South America and Europe. and there is thus a crucial need to collect more measurements in other regions of the world, especially in Africa. In addition, despite the strong correlation between TN and $NO_3^-$ concentrations, the application of the resulting empirical equation (Eq. 30) also adds uncertainties in the observational dataset (Pisani et al., 2017; Niu et al., 2022).

## 4. Conclusions

We developed a global N lateral transfer model from land to oceans through the river network, including the decomposition of DON and PON and denitrification of DIN during fluvial transport. Evaluations using observation data from GRDC and GRQA indicate that ORCHIDEE-NLAT reproduce observed rates and seasonal variations of water discharge and N flow well. The global simulations of ORCHIDEE-NLAT shows that global TN inputs into rivers, TN exports to oceans and denitrification rates in rivers increased significantly over the last century. In particular, the TN export to oceans increased from 27.1 Tg N $yr^{-1}$ during 1901-1910 to 40.8 Tg N $yr^{-1}$ during 2001-2014, with DIN contributing 80% to the TN increase. Our results reveal significant spatial heterogeneity in the global distribution of N inputs, transformation and exports to oceans, with East Asia and Southeast Asia as hotspots of N lateral transfers and their increase. The seasonal amplitude of TN exports are of similar magnitude than the large-scale spatial heterogeneity in TN fluxes. Although the global and continental-scale TN export to oceans simulated by ORCHIDEE-NLAT is similar to that of another widely used model (GlobalNEWS2), their spatial distribution at the basin scale reveals significant discrepancies. One key strength of ORCHIDEE-NLAT is its ability to resolve N processes at the daily timescale, using a framework fully compatible with land surface model outputs, hence allowing to account for the effects of climate change, atmospheric composition changes, land-use change, and agricultural practices (e.g., manure and fertiliser use) in a fully consistent way.

ORCHIDEE-NLAT has however its own limitations and we plan to further enhance its capabilities with additional processes (e.g. autotrophy, variable C:N ratios, erosion-deposition on river bed), additional sources (e.g. aquaculture, direct N deposition) and interconnections with other (semi)-aquatic and benthic systems (hyporheic zone, lakes, reservoirs, floodplains). We will also continue

to collect more observation data to further calibrate and evaluate ORCHIDEE-NLAT. Last but not least, ORCHIDEE-NLAT is currently being dynamically embedded into ORCHIDEE-3 (Vuichard et al., 2019), the land surface scheme embedded in the IPSL Earth System Model opening new avenues towards fully coupled simulations of the land-ocean-atmosphere N cycle. The current offline version of our model could also be easily coupled to other LSMs representing N cycling in terrestrial ecosystems.

857 **Appendices**

858 Table A1. Abbreviation used in the text.

| Abbreviation | Meaning | unit |
|---|---|---|
| $F_{DR_DIN}$ | leaching rates of DIN with drainage | g N d$^{-1}$ |
| $F_{DR_DON}$ | leaching rates of DON with drainage | g N d$^{-1}$ |
| $F_{RO_DIN}$ | leaching rates of DIN with runoff | g N d$^{-1}$ |
| $F_{RO_DON}$ | leaching rates of DON with runoff | g N d$^{-1}$ |
| $F_{RO_PON}$ | erosion rates of PON with runoff | g N d$^{-1}$ |
| $F_{sewage_DIN}$ | DIN inflow rates from sewage | g N d$^{-1}$ |
| $F_{sewage_DON}$ | DON inflow rates from sewage | g N d$^{-1}$ |
| $F_{fastout_H2O}$ | outflow rates of water from fast reservoirs to stream reservoirs | m$^3$ d$^{-1}$ |
| $F_{fastout_DIN}$ | outflow rates of DIN from fast reservoirs to stream reservoirs | g N d$^{-1}$ |
| $F_{fastout_DON}$ | outflow rates of DON from fast reservoirs to stream reservoirs | g N d$^{-1}$ |
| $F_{fastout_PON}$ | outflow rates of PON from fast reservoirs to stream reservoirs | g N d$^{-1}$ |
| $F_{slowout_H2O}$ | outflow rates of water from slow reservoirs to stream reservoirs | m$^3$ d$^{-1}$ |
| $F_{slowout_DIN}$ | outflow rates of DIN from slow reservoirs to stream reservoirs | g N d$^{-1}$ |
| $F_{slowout_DON}$ | outflow rates of DON from slow reservoirs to stream reservoirs | g N d$^{-1}$ |
| $F_{streamout_H2O}$ | outflow rates of $H_2O$ to downstream reservoirs | m$^3$ d$^{-1}$ |
| $F_{streamout_DIN}$ | outflow rates of DIN to downstream reservoirs | g N d$^{-1}$ |
| $F_{streamout_DON}$ | outflow rates of DON to downstream reservoirs | g N d$^{-1}$ |
| $F_{streamout_PON}$ | outflow rates of PON to downstream reservoirs | g N d$^{-1}$ |
| $R_{fast_DIN}$ | denitrification rates in fast reservoirs | g N d$^{-1}$ |
| $R_{fast_DON}$ | decomposition rates of DON in fast reservoirs | g N d$^{-1}$ |
| $R_{fast_PON}$ | decomposition rates of PON in fast reservoirs | g N d$^{-1}$ |
| $R_{slow_DIN}$ | denitrification rates in slow reservoirs | g N d$^{-1}$ |
| $R_{slow_DON}$ | decomposition rates of DON in slow reservoirs | g N d$^{-1}$ |
| $R_{stream_DIN}$ | denitrification rates in stream reservoirs | g N d$^{-1}$ |
| $R_{stream_DON}$ | decomposition rates of DON in stream reservoirs | g N d$^{-1}$ |
| $R_{stream_PON}$ | decomposition rates of PON in stream reservoirs | g N d$^{-1}$ |
| $S_{fast_H2O}$ | water stock in fast reservoir | m$^3$ |
| $S_{fast_DIN}$ | DIN stock in fast reservoir | g N |
| $S_{fast_DON}$ | DON stock in fast reservoir | g N |
| $S_{fast_PON}$ | PON stock in fast reservoir | g N |
| $S_{slow_H2O}$ | water stock in slow reservoir | m$^3$ |
| $S_{slow_DIN}$ | DIN stock in slow reservoir | g N |
| $S_{slow_DON}$ | DON stock in slow reservoir | g N |
| $S_{stream_H2O}$ | water stock in stream reservoir | m$^3$ |
| $S_{stream_DIN}$ | DIN stock in stream reservoir | g N |
| $S_{stream_DON}$ | DON stock in stream reservoir | g N |
| $S_{stream_PON}$ | PON stock in stream reservoir | g N |
| $TW$ | water temperature | °C |
| $F_{T_DIN}$ | dependency of denitrification on temperature | unitless |

| | | |
|---|---|---|
| *depth* | depth of rivers | m |
| *Q* | water discharge | $m^3$ $yr^{-1}$ |

859 Table A2. Values of the key parameters used in ORCHIDEE-NLAT to simulate
860 the lateral transfer of N.

| Parameter | Value | Description | Source |
|---|---|---|---|
| $\tau_{fast}$ | 3.0 days | A factor which translates the topographic index into the water residence time of the “fast” reservoir (Eq. 1) | Ngo-Duc et al., 2006 |
| $\tau_{slow}$ | 25.0 days | A factor which translates the topographic index into the water residence time of the “slow” reservoir (Eq. 2) | Ngo-Duc et al., 2006 |
| $\tau_{stream}$ | 0.24 days | A factor which translates the topographic index into the water residence time of the “stream” reservoir (Eq. 3) | Ngo-Duc et al., 2006 |
| $K_{PON}$ | 0.028 $d^{-1}$ | the average PON decomposition rate at 20℃ in water (Eqs. 20-21) | Islam et al., 2012 |
| $K_{DON}$ | 0.07 $d^{-1}$ | the average DON decomposition rate at 20℃ in water (Eqs. 22-24) | Xia et al., 2013 |
| $K_{DIN}$ | 0.15 $d^{-1}$ | the average denitrification rate in water at 25℃ (Eqs. 25-27) | Alexander et al., 2000 |
| $Q_{10}$ | 2.0 | the temperature sensitivity of PON and DON decomposition rates (Eqs. 20-24) | Liu et al., 2021<br>Zang et al., 2020 |
| $T_{ref1}$ | 20 °C | the reference temperature for PON and DON decomposition (Eqs. 20-24) | |
| $T_{ref2}$ | 25 °C | the reference temperature for denitrification (Eq. 28) | Ma et al., 2022 |

861

**Code and data availability.** The source code of the ORCHIDEE-NLAT model is available online (http://doi.org/10.5281/zenodo.13309551). All forcing and validation data used in this study are publicly available online. The specific sources for these data can be found in Table 1.

**Author contributions.** MM, HZ, RL, PR and PC designed the study. MM and HZ conducted the model development and simulation experiments. PR, RL and PC provided critical contributions to the model development and the design of simulation experiments. MM conducted the model calibration, validation, and data analysis. HZ, PR, RL and PC provided support on collecting forcing and validation data. MM wrote the paper. All authors contributed to interpretation and discussion of results and improved the paper.

**Competing interests.** The contact author has declared that none of the authors has any competing interests.

**Acknowledgements.** MM and PR acknowledge funding from the European Union's Horizon 2020 research and innovation program under grant agreement no. 101003536 (ESM2025 – Earth System Models for the Future). P.R. received financial support from BELSPO through the project ReCAP (which is part of the Belgian research programme FedTwin). HZ acknowledges the Fundamental and Applied Basic Research Fund of Guangdong Province, China (No. 2024A1515010929) and the Fundamental Research Funds for the Central Universities, Sun Yat-sen University (No. 31610004). PC and RL acknowledge support from the CLAND convergence institute funded by the National Research Agency of France 'ANR' 16-CONV-0003. PC also acknowledges support of the CALIPSO project funded through the generosity of Eric and Wendy Schmidt by recommendation of the Schmidt Futures program. RL and

PR further acknowledge funding under the 'France 2030' programme with the reference ANR-22-PEXF-0009 (PEPR 'FairCarboN'—project 'DEEP-C'). We thank Hanqin Tian's team for providing the simulated data from DLEM.

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
