# Peer review of "Estimating lateral nitrogen transfer through the global river network using a land surface model"

_Earth System Dynamics, 2024_

## Referee Comment (RC2)

**Manuscript:** Estimating lateral nitrogen transfer through the global river network using a land surface model

**Major remarks**

The authors developed a new scheme for the lateral transfer of nitrogen over the land surface and via the river network at 0.5° resolution. They implemented their scheme into the land surface model ORCHIDEE and named it ORCHIDEE-NLAT. The scheme considers three nitrogen compounds: PON, DON, and DIN. The manuscript presents an important contribution to Earth System Modelling. It utilizes the ORCHIDEE capabilities by providing daily nitrogen loads, and not only annual loads as in existing previous studies. It also comprises a good discussion on uncertainties (Sect. 3.4)

What I do not understand is why they did not run the full ORCHIDEE model themselves. Instead, ORCHIDEE-NLAT offline scheme was fed by output from ORCHIDEE-CNP and ORCHIDEE-Clateral. Hence, its results heavily rely on input data from other ORCHIDEE versions. If the present offline scheme is an independent model, why it is also called ORCHIDEE? Are there any processes duplicated in ORCHIDEE-NLAT, which had already been simulated by these other ORCHIDEE version? It should be clarified whether specific characteristics of the model output are due to the process representation in ORCHIDEE-NLAT or whether they originate from the used input from the other OCHIDEE versions.

Lateral nitrogen flows are simulated for the period 1901-2014. Unfortunately, no information on the atmospheric forcing for the land surface model is provided (i.e. for the input provided by ORCHIDEE_CNP).

I do not find the evaluation of the model results in Sect. 3.1 to be very convincing. In this respect, Figure 4 shows a rather trivial logarithmic plot where large (low) simulated discharge/N values correspond to large (low) observed values. It shows that the model values are generally of the right order of magnitude but hide the true magnitude of the biases. It may be better to show NSE or RRMSE in such a figure. In this respect, Figure 5 shows large biases with RRSME greater than 30% and medium to low NSE for the three rivers considered. While I do not expect a high performance for nitrogen loads, I am rather surprised by the low performance of the simulated river discharge. If this performance is already low, it will most likely prevent a good performance of the simulated nitrogen loads. In addition, it is implied that Fig. S4 shows a good agreement with the assessment of Marzadri et al. (2021), which I strongly disagree (see comment below). Also, the reasoning for existing model biases is insufficient (see comments below). In my opinion, the evaluation section requires a strong improvement before it is suitable for publication.

Another point of concern is that the paper uses rather short reference periods for comparison (1900-1910, 1991-2000 and 2001-2014). This is too short for climatological studies and the identification of trends, especially given the large interannual and decadal variability in hydrological variables, i.e. precipitation and river runoff, which largely influence the lateral nitrogen flows into the ocean.

As the manuscript includes a lot of typos and some overly long sentences, I recommend a thorough English proof reading.

In summary, the paper describes a relevant model development and provides valuable results, but currently suffers from several flaws, especially in the evaluation section and in the robust identification of trends. Hence, it may be accepted for publication after major revisions are conducted.

**Minor remarks**

In the following suggestions for editorial corrections are marked in *Italic*.

Line 26
I found the naming of the new scheme (ORCHIDE-NLAT) inconsistent with the previously established lateral transfer scheme for carbon (ORCHIDEE-Clateral). In addition, NLAT is a typical abbreviation for No. of latitudes. I suggest a consistent renaming of the new scheme to ORCHIDEE-Nlateral.

Line 182
… of *the model* driving …

Line 201, 214 and 218
In Sect. 2.1.2, you are referring to Table 1 several times. I could hardly find the table until I realized that it is located in Sect. 2.3.1 nine pages later.

Line 213
… and *the data* were downscaled …

Line 219 – Sect. 2.1.3
Sect. 2.1.3 comprises several sets of very similar equations, e.g. eqs.1-3, 4-8, 12-16, 17-19, 20-24, 25-27. This makes this section lengthy and repetitive. Please shorten!

Line 354
… flow rates *are* equal to …

Line 402 and 407
The RPE is commonly defined as mean bias or mean bias error (MBE). Please use one of the two common terms.

Line 403
Please provide the definition of the coefficient of determination that you have used.

Line 426-428
Gramma of sentence seems wrong. Please improve.

Line 439
Evaluation of the *simulated water discharge using* …

Line 447
The unit $m^3/yr$ is strange. Please use of the common units for river discharge: $m^3/s$ or $km^3/yr$.

Line 447-448
It is written:

"…indicating that large errors only occur at some sites draining relatively small basins"

This is not necessarily the case. Such an error may also occur in large basins in dry areas. Please clarify!

Line 449-454
No, there are more factors. A very important factor is actually that biases in the land surface water balance of ORCHIDEE will introduce biases in runoff and, hence, in the discharge. And as you are using runoff inputs from an ORCHIDEE simulation, this factor is very likely the largest factor contributing to biases in streamflow/discharge.

Line 464-465
See comment to line 447-448.

Line 482-484
It is written:
"Nevertheless, the agreement between both assessments (Fig. S4) lends further confidence in the capacity of our model to realistically simulate the N cycle along the global river network."

I strongly disagree with this statement as Fig.S4 indicates considerable differences between both assessments.

Line 513-515
It is written:
"The reality of this transient peak is however questionable as it results mostly from meteorological forcing, which is uncertain for the beginning of the 20th century."

Unfortunately, no information on the atmospheric forcing is provided (see major remarks).

Line 531
… the grid boxes with …

Line 541-545
Sentence is too long and difficult to read. Please rephrase.

Line 596
…simulations, *and* downscale …

Line 595-596
It is written:
"ocean biogeochemical modelling community typically uses annual mean TN fluxes derived from Global News to force their simulations"

Please provide solid reference(s) for this statement, i.e. that is more than a utilization in a single study.

Line 602
… into *rivers*, denitrification …

Line 624-628

Sentence is too long and difficult to read. Please rephrase.

Line 726
… model *used* in …

Line 825
… *reproduces* …

Line 827
… global *simulation* of …

References
The reference section has to be carefully checked as many references include the full names of the authors instead of initials for the given names.

---

## Author Comment (AC1)

**Point-by-point response to the reviewers' comments**

The comments from the reviewers are in bold followed by our responses in regular text. The text in quotation marks represents the content we revised in the new manuscript. And following the reviewer's suggestion, we renamed the model developed in this study from ORCHIDEE_NLAT to LSM_Nlateral_Off.

**■ Reviewer 1**

**This study introduced a newly developed offline model of lateral N transfers, called ORCHIDEE_NLAT, within the framework of the land surface model ORCHIDEE. The ORCHIDEE_NLAT was used to simulate historical changes in riverine DON, PON, and DIN exports across the globe. Overall, it is an important work of global riverine N transport model development. The manuscript is well written, and the model structure is clearly illustrated. Currently, the accuracy of the model in simulating riverine N exports is actually low, especially at regional scale. I understand it is very challenging to accurately simulate N transfers at the global level, but I still have some suggestions for authors to improve the model in the future.**

**➤ Major comments**

1. **The ORCHIDEE-CNP and ORCHIDEE-Clateral are both used to provide the land-to-river inputs. ORCHIDEE-Clateral provides runoff, drainage, DOC, and POC inputs, while ORCHIDEE-CNP provides inputs of DON and DIN leaching from manure. Can ORCHIDEE-CNP provide all these forcing data? Using the outputs of two models may bring uncertainties and make this study complicated. Since runoff and drainage are critical components that determine DIN, DON, and PON fluxes, different water inputs simulated by two versions of land models can bring inconsistencies in water flux information behind N fluxes.**

Thanks for your thoughtful comment on the forcing data of our model. Indeed, it is better to have all input data from the same version of the ORCHIDEE model, compared to from two different versions of the model. However, the dissolved organic matter pool (DOC and DON) and the erosion of particulate organic matter (POC and PON) has not been explicitly represented in ORCHIDEE-CNP. We thus cannot obtain the DOC and DON, nor the POC and PON inputs to the rivers from ORCHIDEE-CNP. ORCHIDEE-

Clateral represents the land-to-river flux of DOC and POC, but does not include a representation of these fluxes for DON and PON, nor of terrestrial N cycling in general. Therefore, we use typical C/N ratios of fluvial dissolved organic matter to estimate PON and DON fluxes from the POC and DOC fluxes simulated with ORCHIDEE-Claretal.

In contrast, ORCHIDEE-CNP represents the leaching of DIN ($NO_3^-$, $NH_4^+$) from soil to the rivers, but not ORCHIDEE-Claretal. Therefore, we used the ORCHIDEE-CNP simulations directly for the DIN inputs to the river networks. To guarantee consistency between DIN inputs from ORCHIDEE-CNP and DOC and POC inputs from ORCHIDEE-Claretal, as well as of the DON and PON inputs estimated from those fluxes, we used the same scheme of soil hydrology (Sun et al., 2021; Zhang et al., 2022) and the same climate forcing data in all simulations. Therefore, the difference in runoff and drainage simulated by these two models are very limited. We will add comparison of global total runoff and drainage between these two models in the supporting information.

2. **In the aquatic N module, why not consider the transformation process from PON to DON, and from inorganic N to organic N?**

We agree that transformation from PON to DON and from inorganic N to organic N are important processes in the ecology of streams and rivers. However, they play a minor role for the flux of total N to the ocean. In our study we aimed to develop a lateral transfer scheme that can be coupled into global land surface models, and which represents the land to ocean transfer of N in a simple but efficient manner. For this, we have ignored some processes that would notably increase the model complexity, but have no strong effect on the simulated riverine exports of total N, and this includes the transformations from PON to DON, and from inorganic N to organic N. Previous studies show that the transformation fraction of riverine POC to DOC during the lateral transport process is limited (about 0.3%) (Zhang et al., 2022). It can thus be inferred that the fraction of PON transformed to DON is also limited and the uncertainties in our results due to ignoring transformation from PON to DON should be limited.

A previous study demonstrated that at global scale river metabolism is strongly dominated by heterotrophic metabolic activities which rely on terrestrial organic matter inputs, whereas in-situ aquatic production only plays a secondary role (Battin et al., 2023). Thus, we assume that also for riverine N cycling, decomposition of organic matter, and denitrification of DIN, is much more important than algae uptake of DIN forming

new PON. Therefore, we have ignored the transformation between inorganic and organic N caused by the growth and mortality of algae. To our knowledge, there is still no reliable model which can well simulate global fluvial autotrophic production of algae and the accompanying N assimilation. Nonetheless, we acknowledge that ignoring the transformation process from PON to DON, and from inorganic N to organic N may result in some uncertainties in our simulation results. We thus have added some text to discuss these potential uncertainties in the revised manuscript.

"The role of autotrophic production is another process currently omitted. Autotrophs (aquatic macrophytes, algae, cyanobacteria, bryophytes, some protists, and bacteria) in freshwater take up DIN from the water column (King et al., 2014) and may play a significant role in river N cycling (Wachholz et al., 2024). In the future, the role of autotrophic production on N retention should thus be considered, although the large dominance of the heterotrophic metabolism on a global scale suggests that in-situ aquatic production is likely a second-order control (Battin et al., 2023)." (lines 830-837)

"The transformation of PON to DON is not included in the current version of LSM_Nlateral_Off. Previous studies show that the transformation fraction of riverine POC to DOC during the lateral transport process is limited (about 0.3%) (Zhang et al., 2022). It can thus be assumed that the fraction of PON transformed to DON is also rather negligible. Nevertheless, we plan to incorporate this transformation process into the LSM_Nlateral_Off model in the next phase of our research." (lines 838-844)

3. **The residence time method is used to calculate N transport along the river networks. This method is commonly used but very simple and may not be able to accurately capture water transport processes. Authors may consider using hydrological kinetic equations in the future.**

Thanks for your excellent advice. Considering the complexity of the hydrodynamic formulas and forcing data required by these formulas, we adopted the residence time method in the first version of the LSM_Nlateral_Off. The residence time method is simple and easy to be applied at a large spatial scale. Nonetheless, as you have suggested, we plan to simulate N transport with the hydrodynamic formulas in the next version of our model. We have proposed this outlook in the revised manuscript. Please see:

"The residence time method was used to calculate water and N transport in the river networks. This method is simple and has been widely used in large scale simulations of fluvial water, carbon and N transports. However, it may not be able to fully capture the seasonality of water and N flow accurately in some areas (Fig. 5 a2 & b2). To more accurately simulate fluvial water and N transports, we plan to replace the residence time method used in current LSM_Nlateral_Off with hydrological kinetic equations in future versions of the model" (Lines 866-873)

4. **The validation of model results only focuses on TN and NO3. (1) How to validate DON and PON flexes? USGS provides organic N observation. (2) The assumption of a linear relationship between observed TN and NO3 may ignore the variations in organic N.**

(1) Thanks for the data information. The USGS provides data on nitrogen concentrations and water discharge across the United States. Based on these data, a previous study (Scott et al., 2007) calculated the long-term (1975–2004) mean annual loads of total organic nitrogen (TON) at 854 stations nationwide. Given that the total nitrogen (TN) flow simulated LSM_Nlateral_Off has been thoroughly evaluated, we assessed the model's performance for specific nitrogen species by comparing the simulated TON fraction with the observed TON fraction reported by Scott et al. (2007). Please see:

"The USGS provides data on N concentrations and water discharge across the United States. Based on these data, a previous study (Scott et al., 2007) calculated the long-term (1975–2004) mean annual loads of total organic N (TON) at 854 stations nationwide. LSM_Nlateral_Off simulates a spatial pattern of TON fraction closely matching that reported by Scott et al. (2007), with high TON fractions in western regions and low fractions in the east (Fig. S3). This suggests that LSM_Nlateral_Off effectively simulates the distribution of specific nitrogen species fractions across the United States, indirectly reflecting the model's reliability." (lines 495-503)

[Figure]

Figure S3. Spatial patterns of long-term (1975–2004) mean annual total organic nitrogen (TON) loads: (a) observed TON fraction reported by Scott et al. (2007); (b) TON fraction simulated by ORCHIDEE-Nlateral.

(2) We agree that the assumption of a linear relationship between observed TN and $NO_3^-$ may ignore the variations in ON. However, previous studies (Romero et al., 2021) and the observation data from GRQA show that there is indeed a significant correlation between TN concentration and $NO_3^-$ concentration (with $R^2$=0.78, Fig. S1). Therefore, the TN calculated from $NO_3^-$ data using an empirical function should be a reasonable choice for evaluating our model at global scale, where TN observation data are still scarce.

5. **The simulated total N fluxes in the 1920s is questionable as authors have already mentioned. What are climate data sources? How about the precipitation change? ORCHIDEE has already been used to simulate lateral C and sediment fluxes, and does the same issue occur in these simulated variables? Better to check it and make it right.**

The climate forcing data is taken from the Global Soil Wetness Project Phase 3 (GSWP3). The forcing data information has been added in Table 1. In the 1920s, the global total amount of heavy rainfall (>25 mm d⁻¹) was higher, which caused more runoff and more TN flow into rivers (Fig. S5). Research of long period runoff fluctuations also reported high global runoff in the 1920s (Probst and Tardy 1989). ORCHIDEE has been used to simulate global lateral C transfer processes, and a similar phenomenon occurs in simulated variables of C lateral transfer (Zhang et al., under review). We have double checked the climate data and model code to make sure our simulations and data analysis are correct. And the relevant expressions are also improved in the revised manuscript:

"The global TN input into rivers, TN exports to oceans and the denitrification in rivers all show a small peak between 1926 and 1931 due to the relatively higher surface runoff during this period (Fig. S4). The reality of this transient peak results mostly from meteorological forcing. During this period, the global total amount of heavy rainfall (>25 mm d⁻¹) was higher, which caused more runoff and more TN flow into rivers (Fig. S5). Research of long period runoff fluctuations also reported high global runoff in during the period (Probst and Tardy 1989)." (lines 549-556)

[Figure]

Figure S5. Global annual TN flow into rivers, runoff and heavy rainfall (> 25 mm d⁻¹) from 1901 to 2014.

**➢ Minor comments**

1. **L64. N leaching into the aquatic environment. LOAC includes land.**

Thanks, "LOAC" has been changed to "aquatic environment".

**2. L91-93. This is also an issue for LSM.**

Sorry for the inaccurate expression. We will change it from

"Therefore, their complexity and high data requirements for calibration and evaluation limit their applicability, in particular the long-term evolution of global N fluxes and transformation processes"

to

"Therefore, their complexity and high requirements for parameterization and forcing datasets limit their applicability, in particular for simulations of the long-term evolution of global N fluxes." (lines 91-93)

**3. L157-159. Does ORCHIDEE-CNP have soil organic C pools? It should also have POC outputs.**

Yes, ORCHIDEE-CNP has three soil organic C pools, physically protected, chemically recalcitrant and active carbon pool. However, ORCHIDEE-CNP does not simulate soil and carbon erosion processes. It thus cannot provide the POC inputs from land to river. Moreover, ORCHIDEE-CNP neither has an explicit DOC pool. Therefore, it cannot simulate the DOC leaching from soils to rivers either.

**4. L166-167. How to separate sewage TN into different N species.**

Sorry for not explaining the separation scheme of sewage in our manuscript. We separated the sewage TN into different N species following Naden et al. (2016), who assume that 10% of sewage TN is DON and the remaining 90% is DIN. We have added this information in the revised manuscript. Please see:

"And the sewage TN was separated into different N species following Naden et al. (2016), who assume that 10% of sewage TN is DON and the remaining 90% is DIN." (lines 222-224)

**5. L197. A constant ratio may make the simulated DON less informative and accurate.**

Based on previous observation, the C:N ratios of dissolved soil organic matter mostly vary within a small range around 12 (8-25). As there is still not a global database on the C:N ratio of DOM for different vegetation types, we have adopted a constant DOC:DON ratio in this study. We acknowledge that the constant DOC:DON ratio may

induce some uncertainties in our simulation results. We have added some text to discuss this issue. Please see:

"The use of a constant C:N ratio to simulate DON fluxes from soils to rivers is another limitation of LSM_Nlateral_Off, since C:N ratio has been shown to vary over time and with land cover type (Li et al., 2019; Yates et al., 2019). The constant C:N may make the estimated DON flux less informative and accurate, and this limitation urgently needs to be overcome in future research." (lines 811-816)

6. **L257. What about N deposition into sediment?**

The PON deposition is mainly affected by the rates of sediment deposition. Sediment and PON deposition are not represented in the current offline model. Recent results from ORCHIDEE-Clateral suggest that about 22% of POC entering river networks is deposited with sediments along the river-floodplain networks (Zhang et al., under review). We have added some text to discuss the potential uncertainties and biases in our simulation results due to the ignorance of PON deposition. Please see:

"At present, few studies have accounted for the effects of PON deposition and resuspension on lateral N transfer in rivers because of the difficulty to represent these processes at the global scale. Moreover, the PON deposition is mainly affected by the rates of sediment deposition which is not represented in the current offline model. Therefore, PON deposition was not simulated yet. Recent results from ORCHIDEE-Clateral suggest that about 22% of POC entering the global river networks is deposited with sediments before reaching the coast (Zhang et al., under review). Assuming a similar fraction of deposited PON, PON export to the oceans simulated by LSM_Nlateral_Off may be approximately 20% lower (about 2 Tg N yr $^{-1}$) than estimated here." (lines 817-826)

7. **Equations 4-11. Not all variables are explained. Please check.**

Thank you for your suggestion. We have added explanations of other variables in the revised manuscript.

"$R_{fast\_PON}$ and $R_{stream\_PON}$ (g N d$^{-1}$) represent PON decomposition rates in the fast and stream reservoirs, respectively. $R_{fast\_DON}$, $R_{slow\_DON}$ and $R_{stream\_DON}$ (g N d$^{-1}$) represent DON decomposition rates in fast, slow and stream reservoirs, respectively. $R_{fast\_DIN}$, $R_{slow\_DIN}$ and $R_{stream\_DIN}$ (g N d$^{-1}$) represent DIN denitrification rates in fast, slow and stream reservoirs, respectively." (lines 297-301)

**8. Figure 4. Put the NSE value into the figure.**

Thanks for your suggestion, we have added the NSE value into Figure 4.

[Figure]

"

Figure 4. Evaluation of LSM_Nlateral_Off. Global-scale comparison between observed and modelled annual-mean water discharge (a) and TN flow (b). Pink symbols represent sites with observations of TN from GRQA, yellow symbols represent GRQA sites for which TN concentrations were estimated from observations of $NO_3^-$, and green symbols represent sites with observations of TN from published literature." (lines 519-525)

**9. L507. Can you explain the decrease in PON export?**

During 1901-2014, the global PON inflow from croplands to the river networks increased because of the increase in cropland area. Meanwhile, the global PON inflow from forest and grassland declined because of the decrease in area and the global greening. Global greening improves vegetation cover (Cortés et al., 2021; Wang et al., 2022) and reduces soil erosion. Altogether, we find a general slight decrease in PON erosion across all plant functional types (PFTs) considered. We have added this explanation in the revised manuscript.

"Global greening enhances vegetation cover (Cortés et al., 2021; Wang et al., 2022) and reduces soil erosion, resulting in a general modest decrease in PON erosion and PON exports to oceans." (lines 548-550)

**10. Figure 7. Better to use mg/L as concentration unit.**

Yes, we have changed the unit to mg/L.

**11. L586. Figs.**

Sorry for the mistake. We have corrected this typo.

**12. L625-627. ON inflow is simulated by ORCHIDEE CNP and Clateral, not NLAT. Right?**

Indeed, the ON inflow from land to rivers is derived from the OC inflow simulated with ORCHIDEE-Clateral and observation-based C:N ratios of dissolved and particulate matter.

**13. Figure 11c. Wrong name of NLAT.**

Sorry for the mistake. We have corrected it.

**14. L738-741. Another important thing is to improve model structure and data quality.**

Indeed, we agree with you, and have added this in the manuscript. Please see:

"This highlights the necessity for improvements in model structure and the quality of both forcing data and evaluation data, as well as the implementation of ensemble-mean assessments, akin to the recent methodologies applied to carbon exports to the oceans (Liu et al., 2024)." (lines 784-787)

**15. L863. Doi is invalid.**

Sorry for the invalid DOI, we uploaded the data and code but forgot to publish it. Now it works, https://zenodo.org/records/13309551.

**■ Reviewer #2**

The authors developed a new scheme for the lateral transfer of nitrogen over the land surface and via the river network at 0.5° resolution. They implemented their scheme into the land surface model ORCHIDEE and named it ORCHIDEE_NLAT. The scheme considers three nitrogen compounds: PON, DON, and DIN. The manuscript presents an important contribution to Earth System Modelling. It utilizes the ORCHIDEE capabilities by providing daily nitrogen loads, and not only annual loads as in existing previous studies. It also comprises a good discussion on uncertainties (Sect. 3.4)

**➤ Major comments**

1. **(1) What I do not understand is why they did not run the full ORCHIDEE model themselves. Instead, ORCHIDEE_NLAT offline scheme was fed by output from ORCHIDEE-CNP and ORCHIDEE-Clateral. Hence, its results heavily rely on input data from other ORCHIDEE versions. (2) If the present offline scheme is an independent model, why it is also called ORCHIDEE? (3) Are there any processes duplicated in ORCHIDEE_NLAT, which had already been simulated by these other ORCHIDEE version? (4) It should be clarified whether specific characteristics of the model output are due to the process representation in ORCHIDEE_NLAT or whether they originate from the used input from the other OCHIDEE versions.**

(1) The overarching idea behind the development of an offline model was to provide a computationally efficient numerical tool in which the mathematical representation of aquatic biogeochemical processes could easily be implemented, calibrated and evaluated. This strategy holds for the processes currently implemented but also for future processes involved in the C-N and other elemental cycles. Furthermore, by construction, it can also be used to route the N leaching fluxes produced by other LSMs (land surface models) in the future, allowing for applications at various scales and across different regions. This has been illustrated in the revised manuscript.

(2) Yes, the present offline scheme is an independent model, and has potential to be forced with output datasets from other land surface models. Therefore, we propose to rename the model LSM_Nlateral_Off (Land Surface Model, N lateral transfer module, Offline).

(3) The water flow processes in LSM_Nlateral_Off are consistent with that in

ORCHIDEE-Claternal, while the lateral N transport processes are newly developed in this study, and have not been simulated in any other versions of ORCHIDEE.

(4) Based on our analysis, the characteristics of water and N inflows from land to rivers are derived from the simulation results of ORCHIDEE-Claternal and ORCHIDEE-CNP. Whereas the characteristics of water and N export to oceans result from the combined influence of the input data from other versions of ORCHIDEE and the process representation in LSM_Nlateral_Off. We have added an explanation on this point in the revised manuscript.

"Input data for LSM_Nlateral_Off are provided by ORCHIDEE-CNP and ORCHIDEE-Claternal, so the characteristics of N inflows from land to rivers are derived from the simulation results of ORCHIDEE-Claternal and ORCHIDEE-CNP. Whereas the characteristics of denitrification and N export to oceans result from the combined influence of the input data from other versions of ORCHIDEE and the process representation in LSM_Nlateral_Off." (lines 531-536)

2. **Lateral nitrogen flows are simulated for the period 1901-2014. Unfortunately, no information on the atmospheric forcing for the land surface model is provided (i.e. for the input provided by ORCHIDEE_CNP).**

Sorry for omitting the information on the climatic forcing data of ORCHIDEE-CNP and ORCHIDEE-Claternal. The climatic forcing datasets for these two models are derived from the Global Soil Wetness Project phase 3 (GSWP3). We have included this information in Table 1. Please see:

"Table 1. List of forcing data of ORCHIDEE-Claternal and ORCHIDEE-CNP and LSM_Nlateral_Off, and the observational data used to evaluate the simulation results. $S_{res}$ and $T_{res}$ are the original spatial and temporal resolution of the forcing data, respectively.

|  | Data | $S_{res}$ | $T_{res}$ | Data source |
| --- | --- | --- | --- | --- |
| Forcing data of ORCHIDEE-Claternal and ORCHIDEE-CNP | Climatic forcing data (precipitation, temperature, incoming shortwave/ longwave radiation, air pressure, wind speed, relative humidity) | 1° | 3 hours | Global Soil Wetness Project Phase 3 (GSWP 3) (Kim et al., 2017) |

| | | | | |
|---|---|---|---|---|
| | Land cover | 0.5° | 1 year | ESA-CCI LUH2v2 database (Hurtt et al., 2011; Lurton et al., 2020) |
| | Soil texture class | 0.5° | – | Reynolds et al. (1999) |
| | Soil bulk density and pH | 30" | – | HWSD v1.2 (FAO/IIASA/ISRIC/ISSCAS/JRC,2012) |
| | Fertilizer application | 0.5° | 1 year | Lu et al., 2017 |
| | Manure application | 5′ | 1 year | Zhang et al., 2017 |
| | Nitrogen deposition | 0.5 | 1 year | IGAC/SPARC CCMI |
| Forcing data of ORCHIDEE-Nlateral | Runoff

Drainage

DOC and POC with runoff

DOC and POC with drainage

Soil temperature (TS) | 1° | 1 day | ORCHIDEE-Clateral

(Zhang et al., 2022) |
| | DIN with runoff and drainage

DON leaching from manure application | 1° | 1 day | ORCHIDEE-CNP

(Sun et al., 2021) |
| | DIN and DON with sewage | 0.5° | 1 year | Beusen et al., 2016 |
| | Flow direction

Topographic index ($f_{topo}$) | 0.5° | / | Vörösmarty et al., 2000 |
| Evaluation data | Riverine water discharge | / | 1 day | GRDC[a] |
| | Riverine TN and $NO_3^-$ concentration | / | point measurement | GRQA[b] |

| | | point | |
|---|---|---|---|
| Riverine TN concentration | / | measurement | Table S1 |

[a] Global Runoff Data Centre (GRDC) (Federal Institute of Hydrology, 2018); [b] Global River water Quality Archive (GRQA) (Virro et al., 2021)." (lines 225-231)

3. **I do not find the evaluation of the model results in Sect. 3.1 to be very convincing. In this respect, Figure 4 shows a rather trivial logarithmic plot where large (low) simulated discharge/N values correspond to large (low) observed values. It shows that the model values are generally of the right order of magnitude but hide the true magnitude of `the biases. It may be better to show NSE or RRMSE in such a figure. (2) In this respect, Figure 5 shows large biases with RRSME greater than 30% and medium to low NSE for the three rivers considered. While I do not expect a high performance for nitrogen loads, I am rather surprised by the low performance of the simulated river discharge. If this performance is already low, it will most likely prevent a good performance of the simulated nitrogen loads. (3) In addition, it is implied that Fig. S4 shows a good agreement with the assessment of Marzadri et al. (2021), which I strongly disagree (see comment below). (4) Also, the reasoning for existing model biases is insufficient (see comments below). In my opinion, the evaluation section requires a strong improvement before it is suitable for publication.**

(1) Thank you for your suggestion. We provide both the logarithmic and normal plots in the revised Fig. 4 and added NSE in the figure. The NSE values indicate that our model accurately captures the observed water and TN flow.

[Figure]

"Figure 4. Evaluation of LSM_Nlateral_Off. Global-scale comparison between observed and modelled annual-mean water discharge (a) and TN flow (b). Pink symbols represent sites with observations of TN from GRQA, yellow symbols represent GRQA sites for which TN concentrations were estimated from observations of $NO_3^-$, and green symbols represent sites with observations of TN from published literature." (lines 519-525)

(2) We agree that the statistics (RRMSE and NSE) for these three sites with long timeseries of nitrogen observations in Fig.5 are not particularly good. However, please note that we performed the model evaluation on a daily time step. It is a big challenge for global-scale river models to accurately simulate water discharge and TN flows at daily time step. The water discharge simulated by ORCHIDEE-Clateral has been evaluated in several previous studies (Lauerwald et al., 2017; Zhang et al., 2022), demonstrating that ORCHIDEE-Clateral effectively simulates both the quantity and seasonality of water discharge. Even so, we will add evaluation of water discharge at other sites around the world where nitrogen observations are not available in support information. Water discharge observations are derived from the GRDC dataset (as used in the original manuscript). Mean bias error (MBE) and the Nash-Sutcliffe efficiency coefficient (NSE) will be applied to assess the performance of LSM_Nlateral_Off in reproducing both the quantity and seasonality of water discharge.

(3) We agree with the reviewer that our paragraph lacked nuance, and we have reformulated the statements, please refer to the response for minor comment #15.

(4) Please refer to the response for minor comment #14.

4. **Another point of concern is that the paper uses rather short reference periods for comparison (1900-1910, 1991-2000 and 2001-2014). This is too short for climatological studies and the identification of trends, especially given the large interannual and decadal variability in hydrological variables, i.e. precipitation and river runoff, which largely influence the lateral nitrogen flows into the ocean.**

Thank you for your valuable advice. In this manuscript, we did not attempt to identify the trends of N variables during the periods of 1901-1910 or 2001-2014. Over the past several decades, the cumulative effects of climate change, increased population, industrialization and agricultural fertiliser use have accelerated the global N cycle, and increased N leaching into river networks (Fowler et al. 2013), which is mainly caused by anthropogenic activities (Beusen et al., 2016). We attempt to quantify the differences in fluxes related to N lateral transfer and transformations between a stage with limited

human influence (1901-1910) and a strong human influence (2001-2014). To avoid the influence of interannual variability in climatological data on N lateral transfer, we employed ten-year average values to compare the differences between the two periods.

For example, global mean annual TN input to rivers during 1901-1910 was 36.81 Tg N yr$^{-1}$, with a standard deviation (SD) of 1.56 Tg N yr$^{-1}$. In contrast, during 2001-2014, the global mean annual TN input is 64.89 Tg N yr$^{-1}$, with a SD of 2.93 Tg N yr$^{-1}$. The interannual variations of global TN inputs to the rivers over 1901-1910 and 2001-2014 are only about 4-5% of the mean annual TN input.

In addition, for the comparison of N inflows simulated by LSM_Nlateral_Off and IMAGE-GNM, we present results for the period from 1901-2000, rather than 1991-2000 in the original manuscript. Please see section 3.3 and Fig. 11 in the original manuscript.

5. **As the manuscript includes a lot of typos and some overly long sentences, I recommend a thorough English proof reading.**

Thank you. We have double checked the whole manuscript and corrected typos and mistakes.

**In summary, the paper describes a relevant model development and provides valuable results, but currently suffers from several flaws, especially in the evaluation section and in the robust identification of trends. Hence, it may be accepted for publication after major revisions are conducted.**

➢ **Minor comments**

**In the following suggestions for editorial corrections are marked in *Italic*.**

1. **Line 26. I found the naming of the new scheme (ORCHIDE-NLAT) inconsistent with the previously established lateral transfer scheme for carbon (ORCHIDEE_ Clateral). In addition, NLAT is a typical abbreviation for No. of latitudes. I suggest a consistent renaming of the new scheme to ORCHIDEE_NLAT.**

Thanks for your suggestion. We have changed ORCHIDEE-NLAT to LSM_Nlateral_Off.

2. **Line 182. "… of *the model* driving …"**

Thanks, we have corrected the text following your suggestion.

3. **Line 201, 214 and 218. In Sect. 2.1.2, you are referring to Table 1 several times. I could hardly find the table until I realized that it is located in Sect. 2.3.1 nine pages later.**

Thanks for your thoughtful suggestion, we have moved Table 1 from Sect. 2.3.1 to Sect. 2.1.2.

4. **Line 213. "… and *the data* were downscaled …"**

Thanks, we have corrected the text following your suggestion.

5. **Line 219. Sect. 2.1.3 comprises several sets of very similar equations, e.g. eqs.1-3, 4-8, 12-16, 17-19, 20- 24, 25-27. This makes this section lengthy and repetitive. Please shorten!**

Thank you for your detailed advice, we have shortened the text by integrating several similar formulas into one formula. For example, Eqs. 1-3 were changed from

$$\text{"}F_{fastout\_H2O} = \frac{S_{fast\_H2O}}{\tau_{fast} \times f_{topo}} \tag{1}$$

$$F_{slowout\_H2O} = \frac{S_{slow\_H2O}}{\tau_{slow} \times f_{topo}} \tag{2}$$

$$F_{downstream\_H2O} = \frac{S_{stream\_H2O}}{\tau_{stream} \times f_{topo}} \tag{3"}$$

to

$$\text{"}F_{out\_H2O} = \frac{S_{H2O}}{\tau \times f_{topo}}$$

where $F_{out\_H2O}$ (km$^3$ d$^{-1}$) represents water outflow rates from fast ($F_{fastout\_H2O}$) /slow ($F_{slowout\_H2O}$) /stream ($F_{downstream\_H2O}$) reservoir; $S_{H2O}$ represents water stock (km$^3$) in fast reservoir ($S_{fast\_H2O}$) /slow reservoir ($S_{slow\_H2O}$) /stream reservoir ($S_{stream\_H2O}$); $\tau$ represents water residence time for each reservoir, 3.0 days, 25.0 days and 0.24 days for fast reservoir, slow reservoir and stream reservoir, respectively (Ngo-Duc et al., 2006); $f_{topo}$ represents grid-cell-specific topographic index (unitless, Vörösmarty et al., 2000)." (lines 246-253)

6. **Line 354. "… flow rates *are* equal to …"**

Thanks, we have corrected the text following your suggestion.

7. **Line 402 and 407. The RPE is commonly defined as mean bias or mean bias error (MBE). Please use one of the two common terms.**

Thanks, we have changed the RPE to mean bias error (MBE).

8. **Line 403. Please provide the definition of the coefficient of determination that you have used.**

Thanks for your suggestion. Since the coefficient of determination is a widely used statistical indicator, we provide a reference that defines it. Please see:

"The $R^2$ represents how much variation in the observations can be explained by the model, for the definition of $R^2$, please refer to Renaud et al. (2010)." (lines 417-419)

9. **Line 426-428. Gramma of sentence seems wrong. Please improve.**

Thanks, we have changed the original text from

"The FV represents the relevant flux, rate or concentration, we have that for each grid cell, the monthly anomaly of FV can be calculated as the difference between the FV value for a given month and its annual mean:"

To

"The FV represents the rate of water flow, denitrification, TN flow rates or TN concentration in rivers. For each grid cell, the monthly anomaly of FV is calculated as the difference between the FV value in a given month and the corresponding annual mean value:" (lines 441-444)

10. **Line 439. "Evaluation of the *simulated water discharge using …*"**

Thanks, we have corrected this sentence following your suggestion.

11. **Line 447. The unit m³/yr is strange. Please use of the common units for river discharge: m³/s or km³/yr.**

Following your suggestion, we have changed the unit from $m^3$ $yr^{-1}$ to $km^3$ $yr^{-1}$.

12. **Line 447-448. It is written**
   **"…indicating that large errors only occur at some sites draining relatively small basins"**

**This is not necessarily the case. Such an error may also occur in large basins in dry areas. Please clarify!**

Thanks for your thoughtful suggestion. The result we intend to show is that the larger errors tend to occur in areas where the water flow is low. We have changed this statement to

"… indicating that large errors tend to occur at sites where water discharge is low" (lines 463-464)

We also have double checked the whole manuscript to make sure all similar errors in the manuscript have been revised.

13. **Line 449-454.**

**No, there are more factors. A very important factor is actually that biases in the land surface water balance of ORCHIDEE will introduce biases in runoff and, hence, in the discharge. And as you are using runoff inputs from an ORCHIDEE simulation, this factor is very likely the largest factor contributing to biases in streamflow/discharge.**

Thanks for your thoughtful suggestion. Uncertainty in runoff and drainage is another important reason causing biases of N lateral transfer. We have added this reason in the revised manuscript as follows:

"(3) the biases in runoff and drainage simulated by ORCHIDEE-Clateral, which may result from deviations in meteorological data and the parameterization of soil hydraulic properties." (lines 470-472)

14. **Line 464-465, see comment to line 447-448.**

Please refer to the detailed response for your minor comments #12.

15. **Line 482-484 It is written:**

**"Nevertheless, the agreement between both assessments (Fig. S4) lends further confidence in the capacity of our model to realistically simulate the N cycle along the global river network."**

**I strongly disagree with this statement as Fig.S4 indicates considerable differences between both assessments.**

We agree with the reviewer that our paragraph lacked nuance, and we have reformulated the statements as follows:

"The simulated DIN concentrations display similar spatial patterns as those obtained from a recent observation-based machine learning (ML) assessment (Marzadri et al., 2021) in regions such as North America, Western Europe, Eastern China, and India. However, in the Amazon, Africa, and Australia, LSM_Nlateral_Off simulated lower DIN concentrations compared to the ML assessment. The lower DIN concentrations in these areas are attributed to different factors. In Australia, low N inflow into rivers results in low DIN concentrations, while in the Amazon and tropical rainforests of Africa, high denitrification rates are primarily responsible for the low DIN concentrations (Fig. 7). The ML involves a significant degree of empirical modelling, and doesn't fully reflect real-world conditions; therefore, this comparison cannot be regarded as a direct evaluation of the model based on data. The consistency between these two models across most regions globally (e.g., North America, Western Europe, Eastern China, and India) suggests that the LSM_Nlateral_Off overall performs reasonably well in simulating DIN lateral transfer processes." (lines 505-519)

16. **Line 513-515 It is written:**

**"The reality of this transient peak is however questionable as it results mostly from meteorological forcing, which is uncertain for the beginning of the 20th century."**

**Unfortunately, no information on the atmospheric forcing is provided (see major remarks).**

The climate forcing data is taken from the Global Soil Wetness Project Phase 3 (GSWP3). The forcing data information has been added in Table 1, please refer to our response for your major comment #2. In the beginning of the 20th century, the global total amount of heavy rainfall (>25 mmin$^{-1}$) was higher, which caused more runoff and more TN flow into rivers (Fig. S5). Research of long period runoff fluctuations also reported high global runoff in the beginning of the 20th century (Probst and Tardy 1989). ORCHIDEE has been used to simulate global lateral C transfer processes, and a similar phenomenon occurs in simulated variables of C lateral transfer (Zhang et al., under review). We have double checked the climate data and model code to make sure our simulations and data analysis are correct. And the relevant expressions are also improved in the revised manuscript:

"The global TN input into rivers, TN exports to oceans and the denitrification in rivers all show a small peak between 1926 and 1931 due to the relatively higher surface runoff during this period (Fig. S4). The reality of this transient peak results mostly from meteorological forcing. During this period, the global total amount of heavy rainfall (>25 mm d$^{-1}$) was higher, which caused more runoff and more TN flow into rivers (Fig. S5). Research of long period runoff fluctuations also reported high global runoff in during the period (Probst and Tardy 1989)." (lines 549-556)

[Figure]

Figure S5. Global annual TN flow into rivers, runoff and heavy rainfall (> 25 mm d$^{-1}$) from 1901 to 2014.

**17. Line 531. "… the grid boxes with …"**

Thanks, we have corrected it.

**18. Line 541-545. Sentence is too long and difficult to read. Please rephrase.**

Thanks, we have rephrased the original sentence from:

"Unsurprisingly, the TN export to oceans increased in most regions since the beginning of the 20th century (Fig. 8c) and in regions such as the south-eastern coastal areas of China, not only the recent TN exports to oceans are relatively high, but also the percentage increase over the 20th century exceeded 100% (Fig. 7c and Fig. 8c)."

to

"Unsurprisingly, TN exports to the oceans have increased in most regions since the early 20th century (Fig. 8c). In regions such as the southeastern coastal areas of

China, TN exports to oceans increased by more than 100% from 1901-1910 to 2001-2014 (Fig. 8c)." (lines 582-586)

19. Line 596. "…simulations, *and* downscale …"

Thanks, we have corrected it.

20. Line 595-596. It is written:

"Ocean biogeochemical modelling community typically uses annual mean TN fluxes derived from Global News to force their simulations"

Please provide solid reference(s) for this statement, i.e. that is more than a utilization in a single study.

Following your suggestion, we have added references for this statement. Please see:

"This result is important because the ocean biogeochemical modelling community typically uses annual mean TN fluxes derived from Global News to force their simulations (Tjiputra et al., 2020)," (lines 620-623)

21. Line 602. "… into *rivers*, denitrification …"

Thanks, we have corrected it.

22. Line 624-628. Sentence is too long and difficult to read. Please rephrase.

Sorry for the lengthy sentence. we have rephased it from:

"The results however markedly differ regarding organic N (ON=PON+DON) with IMAGE-GNM simulating a significant increase from 24.9 Tg N yr$^{-1}$ during 1901-1910 to 37.9 Tg N yr$^{-1}$ in during 1990-2000, while the ON inflow simulated by LSM_Nlateral_Off shows a weaker increasing trend over the same period (26.5 Tg N yr$^{-1}$ during 1901-1910 to 32.4 Tg N yr$^{-1}$ during 1990-2000)."

to

"Therefore, the organic nitrogen (ON = PON + DON) fluxes simulated by IMAGE-GNM and LSM_Nlateral_Off differ significantly. The ON inflow simulated by IMAGE-GNM shows a substantial increase from 24.9 Tg N yr$^{-1}$ during 1901-1910 to 37.9 Tg N yr$^{-1}$ during 1990-2000, while LSM_Nlateral_Off simulates a weaker increasing trend over the same period, from 26.5 Tg N yr$^{-1}$ to 32.4 Tg N yr$^{-1}$." (lines 664-669)

**23.** **Line 726. "… model *used* in …"**

Thanks, we have corrected it.

**24.** **Line 825. "… *reproduces* …"**

Thanks, we have corrected it.

**25.** **Line 827. "… global *simulation* of …"**

Thanks, we have corrected it.

**26.** **References.**
**The reference section has to be carefully checked as many references include the full names of the authors instead of initials for the given names.**

Sorry for the mistakes. we have double checked the reference section to make all references be listed following the guide in the author guide of the ESD.

**References in the response letter**

Battin, T. J., R. Lauerwald, E. S. Bernhardt, E. Bertuzzo, L. G. Gener, R.O. Hall, Erin R. Hotchkiss, et al.: River Ecosystem Metabolism and Carbon Biogeochemistry in a Changing World, Nature, 613, 449–59. https://doi.org/10.1038/s41586-022-05500-8, 2023.

Beusen, A. H. W., A. F. Bouwman, L. P. H. Van Beek, J. M. Mogollón, and J. J. Middelburg.: Global Riverine N and P Transport to Ocean Increased during the 20th Century despite Increased Retention the along Aquatic Continuum, Biogeosciences, 13, 2441–51, https://doi.org/10.5194/bg-13-2441-2016, 2016.

Cortés, J., M. D. Mahecha, M. Reichstein, R. B. Myneni, C. Chen, A. Brenning.: Where Are Global Vegetation Greening and Browning Trends Significant? Geophysical Research Letters, 48(6), e2020GL091496. https://doi.org/10.1029/2020GL091496, 2021.

FAO/IIASA/ISRIC/ISSCAS/JRC. (2012). Harmonized World Soil Database (version 1.2).

Fowler, D., M. Coyle, U. Skiba, M. A. Sutton, J. N. Cape, S. Reis, L. J. Sheppard, et al.: The Global Nitrogen Cycle in the Twenty-First Century, Philosophical Transactions of the Royal Society B: Biological Sciences, 368, 20130164, https://doi.org/10.1098/rstb.2013.0164, 2013.

Hegglin, M. I. et al. IGAC/SPARC Chemistry–Climate Model Initiative (CCMI) 2014 Science Workshop. SPARC Newsl. 43, 32–35, 2014

Hurtt, G. C., L. P. Chini, S. Frolking, et al.: Harmonization of land-use scenarios for the period 1500–2100: 600 years of global gridded annual land-use transitions, wood harvest, and resulting secondary lands. Climatic Change, 109(1-2), 117-161. https://doi:10.1007/s10584-011-0153-2, 2011.

Kim, H.: Global Soil Wetness Project Phase 3 Atmospheric Boundary Conditions (Experiment 1) [Data set], Data Integration and Analysis System (DIAS), https://doi.org/10.20783/DIAS.501, 2017.

King, S. A., J.B. Heffernan, and M. J. Cohen.: Nutrient Flux, Uptake, and Autotrophic Limitation in Streams and Rivers. Freshwater Science, 33, 85–98, https://doi.org/10.1086/674383, 2014.

Lauerwald, R., P. Regnier, M. Camino-Serrano, B. Guenet, M. Guimberteau, A. Ducharne, J. Polcher, and P. Ciais.: ORCHILEAK (Revision 3875): A New Model Branch to Simulate Carbon Transfers along the Terrestrial–Aquatic Continuum of the Amazon Basin. Geoscientific Model Development, 10, 3821–59, https://doi.org/10.5194/gmd-10-3821-2017, 2017.

Li, Mengfan, Jing Wang, Ding Guo, Ruirui Yang, and Hua Fu.: Effect of Land Management Practices on the Concentration of Dissolved Organic Matter in Soil: A Meta-Analysis. Geoderma, 344, 74–81, https://doi.org/10.1016/j.geoderma.2019.03.004, 2019.

Liu, Z., Z. Deng, S. J. Davis, and P. Ciais.: Global Carbon Emissions in 2023. Nature Reviews Earth & Environment, 5, 253–54. https://doi.org/10.1038/s43017-024-00532-2, 2024.

Lu, C, and HQ Tian.: Global nitrogen and phosphorus fertilizer use for agriculture production in the past half century: shifted hot spots and nutrient imbalance. Earth System Science Data, 9(1), 181-192, https://doi.org/10.5194/essd-9-181-2017, 2017.

Lurton, T., Y. Balkanski, V. Bastrikov, S. Bekki, et al.: Implementation of the CMIP6 Forcing Data in the IPSL-CM6A-LR Model. Journal of Advances in Modeling Earth Systems, 12(4). https://doi:10.1029/2019ms001940, 2020.

Marzadri, A., G. Amatulli, D. Tonina, A. Bellin, L. Q. Shen, G. H. Allen, and P. A. Raymond.: Global Riverine Nitrous Oxide Emissions: The Role of Small Streams and Large Rivers. Science of The Total Environment, 776 145148, https://doi.org/10.1016/j.scitotenv.2021.145148, 2021.

Naden, P., V. Bell, E. Carnell, S. Tomlinson, U. Dragosits, J. Chaplow, L. May, E. Tipping: Nutrient fluxes from domestic wastewater: A national-scale historical perspective for the UK 1800–2010. *Science of The Total Environment*, 572, 1471–84, https://doi.org/10.1016/j.scitotenv.2016.02.037, 2016.

Ngo-Duc, T., J. Polcher, and K. Laval.: A 53-Year Forcing Data Set for Land Surface Models. Journal of Geophysical Research: Atmospheres 110, D06116 ,https://doi.org/10.1029/2004JD005434, 2005.

Probst, J.L., Y. Tardy.: Global runoff fluctuations during the last 80 years in relation to world temperature change. American Journal of Science, 289, 267-285, 1989.

Renaud, O., and M. Victoria-Feser.: A Robust Coefficient of Determination for Regression. Journal of Statistical Planning and Inference, 140(7), 1852–62, https://doi.org/10.1016/j.jspi.2010.01.008, 2010.

Reynolds, C., T. Jackson, and W. Rawls.: Estimating available water content by linking the FAO soil map of the world with global soil profile databases and pedo-transfer functions, EOS, Transactions, AGU, Spring Meet. Suppl., 80, S132, https://doi.org/10.1029/2000WR900130, 1999.

Romero, E., W. Ludwig, M. Sadaoui, L Lassaletta, A. F. Bouwman, et al.: The Mediterranean Region as a Paradigm of the Global Decoupling of N and P Between Soils and Freshwaters. Global Biogeochemical Cycles, 35 (3), https://doi.org/10.1029/2020GB006874, 2021.

Scott, D., J. Harvey, R. Alexander, and G. Schwarz.: Dominance of Organic Nitrogen from Headwater Streams to Large Rivers across the Conterminous United States. Global Biogeochemical Cycles, 21(1), https://doi.org/10.1029/2006GB002730, 2007.

Sun, Y., D. S. Goll, J. Chang, P. Ciais, B. Guenet, J. Helfenstein, Y. Huang, et al.: Global Evaluation of the Nutrient-Enabled Version of the Land Surface Model ORCHIDEE-CNP v1.2 (R5986). Geoscientific Model Development, 14, 1987–2010, https://doi.org/10.5194/gmd-14-1987-2021, 2021.

Tjiputra, J. F., J. Schwinger, M. Bentsen, A. L. Morée, S. Gao, I. Bethke, C. Heinze, et al.: Ocean Biogeochemistry in the Norwegian Earth System Model Version 2 (NorESM2). Geoscientific Model Development, 13(5), 2393–2431, https://doi.org/10.5194/gmd-13-2393-2020,2020.

Virro, H., G. Amatulli, A. Kmoch, L. Shen, and E. Uuemaa.: GRQA: Global River Water Quality Archive. Earth System Science Data, 13, 5483–5507, https://doi.org/10.5194/essd-13-5483-2021, 2021.

Vörösmarty, C. J., B. M. Fekete, M. Meybeck, and R. B. Lammers.: Geomorphometric Attributes of the Global System of Rivers at 30-Minute Spatial Resolution. Journal of Hydrology, 237,17–39, https://doi.org/10.1016/S0022-1694(00)00282-1, 2000.

Wachholz, A., J. W. Jawitz, and D. Borchardt.: From Iron Curtain to Green Belt: Shift from Heterotrophic to Autotrophic Nitrogen Retention in the Elbe River over 35 Years of Passive Restoration. Biogeosciences, 21, 3537–50, https://doi.org/10.5194/bg-21-3537-2024, 2024.

Wang, ZQ, H Wang, TF Wang, LN Wang, X Liu, K Zheng, XT Huang: Large discrepancies of global greening: Indication of multi-source remote sensing data. *Global Ecology and Conservation,* 34, e02016. https://doi.org/10.1016/j.gecco.2022.e02016, 2022.

Yates, C. A., P. J. Johnes, A.T. Owen, F. L. Brailsford, H. C. Glanville, C. D. Evans, M. R. Marshall, et al.: Variation in Dissolved Organic Matter (DOM) Stoichiometry in U.K. Freshwaters: Assessing the Influence of Land Cover and Soil C:N Ratio on DOM Composition. Limnology and Oceanography, 64, 2328–40, https://doi.org/10.1002/lno.11186, 2019.

Zhang, B., H. Tian, C. Lu, S. R. S. Dangal, J. Yang, and S. Pan.: Global Manure Nitrogen Production and Application in Cropland during 1860–2014: A 5 Arcmin Gridded Global Dataset for Earth System Modelling. Earth System Science Data, 9(2), 667–78, https://doi.org/10.5194/essd-9-667-2017, 2017.

Zhang, H., R. Lauerwald, P. Regnier, P. Ciais, K. Van Oost, V. Naipal, B. Guenet, and W. Yuan.: Estimating the Lateral Transfer of Organic Carbon through the European River Network Using a Land Surface Model. Earth System Dynamics, 13, 1119–44, https://doi.org/10.5194/esd-13-1119-2022, 2022.

---

## Author Response (AR1)

**Point-by-point response to the reviewers' comments**

The comments from the reviewers are in bold followed by our responses in regular text. The text in quotation marks represents the content we revised in the new manuscript. And following the reviewer's suggestion, we renamed the model developed in this study from ORCHIDEE\_NLAT to LSM\_Nlateral\_Off.

**Reviewer 1**

This study introduced a newly developed offline model of lateral N transfers, called ORCHIDEE\_NLAT, within the framework of the land surface model ORCHIDEE. The ORCHIDEE\_NLAT was used to simulate historical changes in riverine DON, PON, and DIN exports across the globe. Overall, it is an important work of global riverine N transport model development. The manuscript is well written, and the model structure is clearly illustrated. Currently, the accuracy of the model in simulating riverine N exports is actually low, especially at regional scale. I understand it is very challenging to accurately simulate N transfers at the global level, but I still have some suggestions for authors to improve the model in the future.

- > Major comments
- 1. The ORCHIDEE-CNP and ORCHIDEE-Clateral are both used to provide the landto-river inputs. ORCHIDEE-Clateral provides runoff, drainage, DOC, and POC inputs, while ORCHIDEE-CNP provides inputs of DON and DIN leaching from manure. Can ORCHIDEE-CNP provide all these forcing data? Using the outputs of two models may bring uncertainties and make this study complicated. Since runoff and drainage are critical components that determine DIN, DON, and PON fluxes, different water inputs simulated by two versions of land models can bring inconsistencies in water flux information behind N fluxes.

Thanks for your thoughtful comment on the forcing data of our model. Indeed, it is better to have all input data from the same version of the ORCHIDEE model than from two different model versions. However, the leaching of dissolved organic matter (DOC and DON) and the erosion of particulate organic matter (POC and PON) are currently not represented in ORCHIDEE-CNP. We thus cannot obtain the DOC and DON, nor the POC and PON inputs to rivers from ORCHIDEE-CNP, this model version only providing the inorganic N fluxes (leaching of DIN ( $NO_3^-$ ,  $NH_4^+$ ) from soil to rivers). Conversely, ORCHIDEE-Clateral represents the land-to-river flux of DOC and POC, yet does not include a representation of the corresponding DON and PON fluxes, nor terrestrial N cycling in general. Therefore, we use typical C/N ratios of fluvial organic matter to estimate PON and DON fluxes from the POC and DOC fluxes simulated by ORCHIDEE-Clateral.

The difference in runoff and drainage simulated by ORCHIDEE-CNP and ORCHIDEE-Clateral are very limited (Fig. S1), as these two models use the same hydrological module (Sun et al., 2021; Zhang et al., 2022) and have been run with the same climate and land use forcing data (Table 1). This clarification has now been included in the revised manuscript.

"Runoff and drainage are critical components that determine DIN, DON, and PON fluxes. As ORCHIDEE-CNP and ORCHIDEE-Clateral used the same scheme to simulate soil hydrology (Sun et al., 2021; Zhang et al., 2022) and they have been run with the same climate and land use forcing data (Table 1).Therefore, the difference in runoff (0.9%) and drainage (1.7%) simulated by the two ORCHIDEE branches are very limited (Fig. S1)." (lines 208-214)

Figure S1. Comparison of global runoff (a) and drainage (b) simulated by ORCHIDEE-Clateral and ORCHIDEE-CNP.

**2. In the aquatic N module, why not consider the transformation process from PON to DON, and from inorganic N to organic N?**

We agree that transformation from PON to DON and from inorganic N to organic N are important processes for the ecology of streams and rivers. However, they play a minor role in the flux of total N to the ocean. In our study, we aimed to develop a lateral transfer scheme that can be coupled to global land surface models, representing the transfer of N from the land to ocean in a simple but efficient manner. For this, we have ignored some processes that would notably increase the model complexity, but have no strong effect on the simulated riverine exports of total N, and these include the transformations from PON to DON, and from inorganic N to organic N. A previous study shows that the transformation fraction of riverine POC to DOC during the lateral transport process is limited (about 0.3%) (Zhang et al., 2022). It can thus be inferred that the fraction of PON to DON is also limited, suggesting that uncertainties due to the omission of PON to DON transformations are limited.

Previous research also found that at global scale river metabolism is strongly dominated by heterotrophic metabolic activities fuelling on terrestrial organic matter inputs, whereas in-situ aquatic production only plays a secondary role (Battin et al., 2023). Thus, we have assumed that for riverine N cycling, decomposition of organic matter and denitrification of DIN are much more important than algae uptake of DIN forming new PON, hence we have ignored the transformation between inorganic and organic N caused by the growth and mortality of algae. To our knowledge, there is still no reliable global model that can well simulate global fluvial autotrophic production of algae and the accompanying N assimilation. Nonetheless, we acknowledge that ignoring the transformation process from PON to DON, and from inorganic N to organic N may result in uncertainties in our simulation results. We have added some text to discuss these potential uncertainties in the revised manuscript.

"The role of autotrophic production is another process currently omitted. Autotrophs (aquatic macrophytes, algae, cyanobacteria, bryophytes, some protists, and bacteria) in freshwater systems take up DIN from the water column (King et al., 2014) and may play a significant role in N cycling within rivers (Wachholz et al., 2024). In future model developments, the role of autotrophic production on N retention should thus be considered, although the large dominance of the heterotrophic metabolism on a global scale suggests that in-situ aquatic production is a second-order control on N cycling (Battin et al., 2023). The transformation of PON to DON is also not included in the current version of LSM\_Nlateral\_Off. A previous study suggests that the instream transformation of POC to DOC is limited (about 0.3%) (Zhang et al., 2022). It can thus be assumed that the fraction of PON transformed to DON is also rather negligible. Nevertheless, we plan to incorporate this transformation process into our model in the next phase of our research." (lines 901-914)

3. The residence time method is used to calculate N transport along the river networks. This method is commonly used but very simple and may not be able to accurately capture water transport processes. Authors may consider using hydrological kinetic equations in the future.

Thanks for your excellent advice. Considering the complexity of the hydrodynamic formulas and the associated forcing data required for their implementation, we adopted the residence time method in the first version of LSM\_Nlateral\_Off. The residence time method is simple and easy to be applied at a large spatial scale. Nonetheless, as you have suggested, simulating N transport with hydrodynamic formulas in the next version of our model is a valuable suggestion for future research and has been included in the outlook section of our revised manuscript:

"The residence time method was used to estimate water and N transport within river networks. This method is simple and has been widely used in large scale simulations of fluvial water, carbon and N transports (Beusen et al., 2015; Jepsen et al., 2019; Zhang et al, 2022). However, it may not fully capture the seasonality of water and N flows accurately in some regions (Fig. 5 a2 & b2). To improve the accuracy of simulating fluvial water and N transport, the residence time method currently used in LSM\_Nlateral\_Off could be replaced with hydrological kinetic equations in future versions of the model." (Lines 874-882)

4. The validation of model results only focuses on TN and NO3. (1) How to validate DON and PON flexes? USGS provides organic N observation. (2) The assumption of a linear relationship between observed TN and NO3 may ignore the variations in organic N.

(1) Thanks for the data information. Indeed, USGS provides data on nitrogen concentrations and water discharge across the United States. Based on these data, a previous study (Scott et al., 2007) calculated the long-term (1975-2004) mean annual loads and fractions of total organic nitrogen (TON) at 854 stations nationwide. Given that the total nitrogen (TN) flow simulated by LSM\_Nlateral\_Off has been thoroughly evaluated, we specifically assessed the model's performance for organic nitrogen by

comparing the simulated TON fraction (i.e. TON yield / TN yield) with the observed TON fraction reported by Scott et al. (2007). Please see:

"As an additional evaluation, we compared our model results against observed N concentrations and water discharges across the United States provided by the U.S. Geological Survey (USGS). Based on these data, a previous study (Scott et al., 2007) calculated the long-term (1975-2004) mean annual loads of total organic N (TON) and TON fractions (TON yield / TN yield) at 854 stations nationwide. LSM\_Nlateral\_Off simulates a spatial pattern for the TON fraction which closely matches that reported by Scott et al. (2007), with high values in western regions and low values in the east (Fig. S7). This suggests that LSM\_Nlateral\_Off not only effectively simulates TN fluxes, but also captures the organic and inorganic fractions across the United States relatively well." (lines 545-555)

---

## Referee Report (RR1)

**Manuscript:** Estimating lateral nitrogen transfer through the global river network using a land surface model

**Major remarks**

Reviewer 2 – Major remark 4
It seems that you missunderstood my comment. May be I should have phrased my comment clearer. I was not talking about trends within a 10 years time period, but about that using only a ten-years mean as a reference may be missleading when comparing two 10-year means to determine the differences (i.e. trend) between these two climates. In hydrology, there is substantial decadal variability with wet and dry decades. Hence, if you accidentally compare a wet decade with a dry decade, you will find a difference (i.e. trend) that is not real just because of decadal variability. Usually, to investigate climatological relevant differences in hydrology, you have to compare at least 30-year means to get robust results.

In general, the authors responded well to the reviewers' comments. After adequately addressing one major remark and some minor remarks, the manuscript may be accepted for publication.

**Minor remarks**

In the following suggestions for editorial corrections are marked in *Italic*.

Sect. 2.1.2

Please note the meteorological foring data (i.e. GSWP3) and the time period that has been used to simulate the runoff and drainage data. This is relevant information the reader should be able to get directly from the text without searching for it in Table 1.

Line 209-211
 ORCHIEE-CNP … …2022)*, and* they …

Line 212-214
Therefore, the *differences* in … …are *relatively small*.

Line 254
… temporal *resolutions* of …

Reviewer 2 - Minor comment 5
Please be more thorough with shortenting unnecessary repetitions. Eq. 2-6 should still be merged into one equation, the same applies to eq. 10-12, and eq. 13-14

Line 380
*Tables* A1 and A2 *provide* a …

Line 516-517
… water *discharge (Fig.* S3b).

Line 589
In *the following*, we …

Line 693-700
It is written:
"Our results indicate that the spatial pattern of seasonal amplitudes in TN concentrations at river mouths differs from that of TN exports (Fig. 10c, d). This result is important because the ocean biogeochemical modelling community typically uses annual mean TN fluxes derived from Global News to force their simulations (e.g., …), and downscales these inputs to monthly values under the assumption that the seasonal variability of the flux is entirely driven by river discharge."

This statement is not very clear to me. Using the approach described, the community assumes constant TN concentrations $C_N$ that are applied to calculate TN exports at the river mouth by multiplying the river discharge $Q$ by $C_N$. However, then the seasonal amplitude of $C_N$ (constant = zero amplitude) is also different to the one of the TN exports (seasonal amplitude induced by the discharge). In my opinion, to have a valid criticism of the common community method, you have to show that the seasonal amplitude of TN exports is significantly different to the seasonal amplitude of river discharge.

Line 712
… (b) *rates of denitrification;* (c) *TN exports to oceans;* …

---

## Author Response (AR2)

**Point-by-point response to the reviewers' comments**

The comments from the reviewers are in bold followed by our responses in regular text. The text in quotation marks represents the content we revised in the new manuscript.

**Major comments**

**Reviewer 2 – Major remark 4**

It seems that you misunderstood my comment. Maybe I should have phrased my comment clearer. I was not talking about trends within a 10years time period, but about that using only a ten-years mean as a reference may be misleading when comparing two 10-year means to determine the differences (i.e. trend) between these two climates. In hydrology, there is substantial decadal variability with wet and dry decades. Hence, if you accidentally compare a wet decade with a dry decade, you will find a difference (i.e. trend) that is not real just because of decadal variability. Usually, to investigate climatological relevant differences in hydrology, you have to compare at least 30-year means to get robust results.

Thank you for your detailed explanation. According to your suggestion and considering the rapid increase of nitrogen (N) fluxes since 1960, we used the 20-year average N fluxes of 1901-1920 and 1995-2014 to quantify the changes from early 20th century to the contemporary period (Figs 8, S10 b). We also analysed the contemporary spatial patterns of water discharges, N fluxes and N concentrations using a 20-years average from 1995 to 2014 (Figs 7, 10, S10 a, S11, and S12). With reference to the IPCC visual guide, we have adjusted the color scheme of the figures above to make them more friendly to color-blind individuals. The relevant content has been updated in the manuscript, mainly in section 3.2 "Temporal and spatial patterns of N flows".

"Averaged over the 1995-2014 period, the annual TN input from soils to rivers, TN exports to oceans and denitrification in transit amount to 64.4 Tg N yr-1, 40.0 Tg N yr-1, and 24.4 Tg N yr-1, respectively. These three N fluxes show increasing trends from 1901 to 2014. The global annual TN input to rivers increased by 72.4 %, from 37.4 Tg N yr-1 during 1901-1920 to 64.4 Tg N yr-1 during 1995-2014 (Fig. 6 a). The global annual TN export to oceans increased by 45.6 % from 27.4 Tg N yr-1 to 40.0 Tg N yr-1. Most of this increase is attributed to DIN, which doubled over the simulation period, rising from 10.0 Tg N yr-1 to 19.9 Tg N yr-1, while, in absolute terms, DON exports show a much smaller increase but still substantial relative increase of 50.6 % (Fig. 6b). In contrast, PON exports to oceans show a slightly decreasing trend. This decrease is mainly attributed to global greening, which enhances vegetation cover (Cortés et al., 2021; Wang et al., 2022) and reduces soil erosion, resulting in lower PON inputs from the land and, thus, PON exports to oceans. The increase in global denitrification mostly follows the rise in DIN inputs, with a relative increase of 146.6 %, from 9.9 Tg N yr-1 during 1901-1920 to 24.4 Tg N yr-1 during 1995-2014 (Fig. 6a)." (lines 601-617)

Figure 7. Spatial patterns of annual mean N fluxes and concentrations during 1995-2014: (a) TN inputs into rivers; (b) denitrification rates in rivers; (c) TN exports to oceans; (d) TN concentrations at rivers mouths. To display the spatial

patterns of denitrification in rivers better, we excluded data with denitrification rates less than 0.001 GN yr-1 per grid." (lines 678-682)

---

## Author Response (AR3)

**Point-by-point response to the editor's comments**

Thank you very much for the thorough revision of the manuscript. All reviewer comments were addressed appropriately, and the manuscript has been revised accordingly. One point, however, remains unclear: what was the rationale for selecting a 20-year period for the comparison of time intervals instead of the 30-year period suggested by the reviewer? A brief clarification on this choice would be helpful to ensure full transparency and strengthen the methodological justification.

Thanks for your thoughtful advice, and we apologize for not clearly explaining the rationales behind selecting a 20-year period. Our goal is to quantify the differences in nitrogen (N) flows related to lateral transfer and transformation between a period with limited human influence and one with strong human impact. Our results (Fig. 6), along with estimates from other models (Beusen et al., 2016a; Seitzinger et al., 2010), indicate that N flows related to lateral transfer and transformation have increased rapidly since 1960. This increase is primarily driven by anthropogenic activities (e.g., fertilization, manure application and sewage) rather than climate change (Beusen et al., 2016; Yao et al., 2020; Van Meter et al., 2018). As shown in Fig. 6, N flows related to lateral transfer and transformation exhibited a significant increased trend during 1985-2014. Using an overly long averaging period to assess N flow changes would dampen the magnitude of these changes. Considering these factors above and the reviewer's suggestions, we chose a 20-year period for the comparison of time intervals instead of the 30-year period suggested by the reviewer.

We have added relevant explanation in section 3.3.2.

"In this section, we examine the spatial distribution of contemporary N flows, and their changes compared to the early 20th century. Given the rapid increase in N flows since 1960 and the interannual variability induced by climate, we use the 1995-2014 average to represent contemporary N flows and the 1901-1920 average to represent early 20th-century conditions." (Lines 632-636)

[Figure]

Figure 6. Trends in global N flows from 1901 to 2014: (a) yearly mean TN inputs into rivers, TN exports to oceans and denitrification rates; (b) yearly mean DIN, DON and PON exports to oceans. TN: total nitrogen; DIN: dissolved inorganic nitrogen; DON: dissolved organic nitrogen; PON: particulate organic nitrogen.

**References**

Van Meter, K. J., Basu, N. B., Veenstra, J. J., and Burras, C. L.: The nitrogen legacy: emerging evidence of nitrogen accumulation in anthropogenic landscapes, Environmental Research Letters, 11 (035014), https://doi.org/10.1088/1748-9326/11/3/035014, 2016.

Beusen, A. H. W., Bouwman, A. F., Van Beek, L. P. H., Mogollón, J. M., and Middelburg, J. J.: Global Riverine N and P Transport to Ocean Increased during the 20th Century despite Increased Retention the along Aquatic Continuum, Biogeosciences, 13, 2441–51, https://doi.org/10.5194/bg-13-2441-2016, 2016a.

Seitzinger, S. P., Mayorga, E., Bouwman, A. F., Kroeze, C., Beusen, A. H. W., Billen, G., Drecht, G. V., et al.: Global River Nutrient Export: A Scenario Analysis of Past and Future Trends, Global Biogeochemical Cycles, 24, GB0A08, https://doi.org/10.1029/2009GB003587, 2010.

Yao, Y., Tian, H., Shi, H., Pan, S., Xu, R., Pan, N. and Canadell, J. G.: Increased Global Nitrous Oxide Emissions from Streams and Rivers in the Anthropocene, Nature Climate Change, 10, 138–42. https://doi.org/10.1038/s41558-019-0665-8, 2020.